# Error as Signal: Stiffness-Aware Diffusion Sampling via Embedded Runge-Kutta Guidance

**Inho Kong**[1*]**, Sojin Lee**[2*]**, Youngjoon Hong**[3,4†]**, Hyunwoo J. Kim**[2†]
[1]Korea University [2]KAIST [3]Seoul National University [4]Korea Institute for Advanced Study
`inh212@korea.ac.kr`
`{sojin.lee, hyunwoojkim}@kaist.ac.kr`
`hongyj@snu.ac.kr`

## ABSTRACT

Classifier-Free Guidance (CFG) has established the foundation for guidance mechanisms in diffusion models, showing that well-designed guidance proxies significantly improve conditional generation and sample quality. Autoguidance (AG) has extended this idea, but it relies on an auxiliary network and leaves solver-induced errors unaddressed. In stiff regions, the ODE trajectory changes sharply, where local truncation error (LTE) becomes a critical factor that deteriorates sample quality. Our key observation is that these errors align with the dominant eigenvector, motivating us to leverage the solver-induced error as a guidance signal. We propose **E**mbedded **R**unge–**K**utta **Guid**ance (ERK-Guid), which exploits detected stiffness to reduce LTE and stabilize sampling. We theoretically and empirically analyze stiffness and eigenvector estimators with solver errors to motivate the design of ERK-Guid. Our experiments on both synthetic datasets and the popular benchmark dataset, ImageNet, demonstrate that ERK-Guid consistently outperforms state-of-the-art methods. Code is available at
`https://github.com/mlvlab/ERK-Guid`.

## 1 INTRODUCTION

Generative models Kingma & Welling (2014); Rezende & Mohamed (2015); Heusel et al. (2017); Lipman et al. (2023); Ho et al. (2020); Song et al. (2021b); Goodfellow et al. (2020) aim to approximate complex data distributions and generate new samples, enabling a wide range of applications in image synthesis Brock et al. (2019); Rombach et al. (2022); Zhang et al. (2023); Kawar et al. (2023); Park et al. (2024b; 2025), editing Brooks et al. (2023), and video generation Gupta et al. (2024); Park et al. (2024a). Among them, diffusion models Ho et al. (2020); Song et al. (2021b); Karras et al. (2022) have emerged as a dominant paradigm, achieving strong performance across diverse generation tasks Gupta et al. (2024); Karras et al. (2024a); Lee et al. (2024); Ko et al. (2024). They define a forward process that gradually perturbs data into Gaussian noise, while a neural network is trained to predict the score function of each noisy distribution. This score estimate parameterizes the reverse-time dynamics, allowing data to be reconstructed through iterative denoising. Sampling is commonly formulated as solving an ordinary differential equation (ODE) or stochastic differential equation (SDE), where the drift is defined by the learned network Karras et al. (2022); Song et al. (2021a); Lu et al. (2022). As a result, the quality of generated samples depends not only on model accuracy but also on the numerical solver used to approximate the reverse dynamics, which can significantly influence fidelity and stability.

Guidance mechanisms emerged to improve both sampling fidelity and perceptual quality by introducing suitable proxies for steering the sampling trajectory. The de facto standard, Classifier-Free Guidance (CFG) Ho & Salimans (2022), combines unconditional and conditional predictions to

---

* Equal contribution. † Corresponding authors.

strengthen alignment and enhance image quality. Predictor-Corrector Guidance (PCG) Bradley & Nakkiran (2024) further refines this view by interpreting CFG as a predictor–corrector update that extrapolates between these predictions. Autoguidance (AG) Karras et al. (2024a) follows a similar principle by contrasting outputs from models of different capacities, using their discrepancies to identify regions where model-induced errors are significant. However, subsequent methods Kynkäänniemi et al. (2024); Sadat et al. (2024); Zheng & Lan (2024); Zhao et al. (2025) rely solely on such model-based differences, overlooking the numerical errors arising from the solver itself as potential guidance signals. We observe that, in stiff regions of the diffusion ODE, the solver's local truncation error (LTE) aligns with the dominant eigenvector of the drift's Jacobian, revealing a numerically grounded proxy distinct from model-space signals and motivating our approach.

In this work, we propose **E**mbedded **R**unge–**K**utta **Guid**ance (**ERK-Guid**), a novel approach that mitigates solver-induced local truncation error (LTE) during diffusion sampling by estimating the dominant eigenvector in stiff regions. In diffusion ODEs, stiffness arises when drift directions change rapidly, and we observe that the resulting LTE consistently aligns with the dominant eigenvector under such conditions. To exploit this property, we introduce two cost-free estimators derived from the Embedded Runge–Kutta (ERK) formulation: *ERK solution difference* (between Heun and Euler solutions) and *ERK drift difference* (between their corresponding drifts). The ratio of their norms serves as **a stiffness estimator** to identify regions of high stiffness, while the ERK drift difference further provides **an eigenvector estimator** to approximate the dominant eigenvector direction. ERK-Guid transforms theoretical insights on stiffness and dominant eigenvector alignment into a practical guidance mechanism. Ultimately, ERK-Guid provides a stable, cost-free proxy that effectively reduces solver-induced errors by applying guidance along this estimated eigenvector direction.

Our contributions are summarized as follows:

- We introduce the Embedded Runge-Kutta Guidance (ERK-Guid), a stiffness-aware guidance method that leverages solver errors as informative signals for diffusion sampling.
- We propose cost-free estimators for stiffness detection and dominant eigenvector estimation, derived from ERK solution and drift differences to determine the guidance direction.
- We design a stabilized guidance scheme that bridges theoretical insights with practical robustness, ensuring robustness without additional network evaluations.
- We demonstrate that ERK-Guid provides an orthogonal guidance signal, integrates into Runge-Kutta-based solvers as a plug-and-play module, and improves upon strong baselines.

## 2 RELATED WORKS

**ODE solvers in diffusion models.** A major research trend in diffusion models has been to accelerate sampling by improving ODE solvers that approximate the underlying probability flow dynamics more efficiently. Early work such as DDIM Song et al. (2021a) reinterprets the stochastic sampling process of DDPM Ho et al. (2020) as a deterministic ODE trajectory, enabling significantly fewer sampling steps without retraining. Building on this view, PNDM Liu et al. (2022) introduces a pseudo-numerical multistep method that generalizes DDIM beyond first-order updates. Subsequent studies further leverage the structure of diffusion dynamics. DEIS Zhang & Chen (2023) employs exponential integrators to reduce discretization error, DPM-Solver Lu et al. (2022) derives high-order solvers with coefficients designed to minimize local truncation error, and UniPC Zhao et al. (2023) unifies predictor–corrector schemes under a single framework.

More recently, DPM-Solver-v3 Zheng et al. (2023) incorporates empirical model statistics into solver parameterization to jointly address discretization and model approximation errors. Bespoke solver approaches develop customized solver designs for a fixed pre-trained model, encompassing methods that optimize time-step schedules Xue et al. (2024) and solver parameters Wang et al. (2025); Shaul et al. (2024). In contrast to these approaches, which redesign or replace the ODE solver, ERK-Guid operates in a fundamentally different regime. We keep the solver fixed and instead leverage the solver's own error: the discrepancy between low- and high-order solver updates. By using this discrepancy as a directional correction, ERK-Guid provides solver-aware guidance without modifying the solver's numerical structure.

**Adaptive guidance computation.** In diffusion models, Classifier-Free Guidance (CFG) Ho & Salimans (2022) has become the de facto standard for improving fidelity and condition alignment by contrasting conditional and unconditional denoisers. Despite its success, CFG often suffers from overshoot, loss of diversity, and entangled fidelity–diversity trade-offs, limiting its flexibility across noise levels. Autoguidance (AG) Karras et al. (2024a) improves robustness by replacing the unconditional branch with a weaker model, correcting model-induced errors without sacrificing variation. Beyond these canonical approaches, several adaptive guidance strategies have been explored. Guidance Interval Kynkäänniemi et al. (2024) activates guidance only at mid-range noise levels. Other works Sadat et al. (2024); Zheng & Lan (2024) mitigate oversaturation under strong guidance, and DyDiT Zhao et al. (2025) dynamically adjusts model capacity across timesteps. These advances demonstrate how CFG has shaped a broad family of model-based guidance mechanisms. In contrast, ERK-Guid employs a solver-driven proxy derived from ERK discrepancies, yielding an orthogonal guidance signal yet complementary to model-based guidance.

## 3 PRELIMINARIES

**Denoising Diffusion Models.** Denoising diffusion models Ho et al. (2020); Song et al. (2021b); Karras et al. (2022) generate samples by simulating the reverse-time dynamics of a predefined stochastic differential equation (SDE). The SDE gradually transforms the data distribution $p_{\text{data}}$ into a perturbed distribution $p(\mathbf{x}; \sigma)$. Following EDM2 Karras et al. (2024b), the perturbed distribution is defined as the convolution of $p_{\text{data}}$ with Gaussian noise Kynkäänniemi et al. (2024), i.e., $p(\mathbf{x}; \sigma) = p_{\text{data}}(\mathbf{x}) * \mathcal{N}(\mathbf{x}; \mathbf{0}, \sigma^2 \mathbf{I})$, where $\sigma \in [0, \sigma_{\max}]$.

The reverse-time SDE can be equivalently reformulated as an ordinary differential equation (ODE) Song et al. (2021b), leading to a deterministic sampling process $\mathbf{x}_0 \sim p_{\text{data}}$, that solves the following initial value problem:

$$\frac{d\mathbf{x}_\sigma}{d\sigma} = \boldsymbol{f}(\mathbf{x}_\sigma; \sigma) = -\sigma \nabla_{\mathbf{x}_\sigma} \log p(\mathbf{x}_\sigma; \sigma), \quad \mathbf{x}_0 = \mathbf{x}_{\sigma_{\max}} + \int_{\sigma_{\max}}^0 \left( \frac{d\mathbf{x}_\sigma}{d\sigma} \right) d\sigma, \tag{1}$$

where $\boldsymbol{f}(\mathbf{x}_\sigma; \sigma)$ denotes the drift function of the ODE, and $\mathbf{x}_\sigma$ refers to the trajectory of the sample as a function of the noise level $\sigma$. The drift function is typically approximated by the learned model $\boldsymbol{f}_\theta(\mathbf{x}_\sigma; \sigma)$, which is trained via score-matching objectives Song et al. (2021b). Note that the initial state $\mathbf{x}_{\sigma_{\max}}$ can be approximately sampled from $\mathcal{N}(\mathbf{0}, \sigma_{\max}^2 \mathbf{I})$ when $\sigma_{\max}$ is sufficiently large.

In practice, the ODE cannot be solved analytically; numerical solvers are employed. To enable numerical integration, EDM2 Karras et al. (2024b) discretizes the interval into a sequence of noise levels $\{\sigma_0, \ldots, \sigma_N\}$, where $N$ is the total number of integration steps. Note that $[\sigma_i, \sigma_{i+1}]$ denotes the $i$-th integration interval, with $\sigma_0 = \sigma_{\max}$ and $\sigma_N = 0$. A numerical solver is then applied to each interval to approximate the following integration:

$$\mathbf{x}_{\sigma_{i+1}} = \mathbf{x}_{\sigma_i} + \int_{\sigma_i}^{\sigma_{i+1}} \boldsymbol{f}(\mathbf{x}_\sigma; \sigma) d\sigma. \tag{2}$$

The Euler method Hairer et al. (1993), a widely used first-order solver, provides the following update and local truncation error (LTE):

$$\mathbf{x}_{\sigma_{i+1}}^{\text{Euler}} = \mathbf{x}_{\sigma_i} - h \boldsymbol{f}(\mathbf{x}_{\sigma_i}; \sigma_i), \tag{3}$$

$$\text{LTE}^{\text{Euler}} = \mathbf{x}_{\sigma_{i+1}} - \mathbf{x}_{\sigma_{i+1}}^{\text{Euler}} = \mathcal{O}(h^2), \tag{4}$$

where $h = \sigma_i - \sigma_{i+1} > 0$ refers the step size. To reduce the local truncation error, higher-order solvers are commonly used. Heun's method Hairer et al. (1993), based on the trapezoidal rule Hairer et al. (1993), introduces a correction to the Euler estimate and effectively incorporates implicit integration. Its update and LTE are given as follows:

$$\mathbf{x}_{\sigma_{i+1}}^{\text{Heun}} = \mathbf{x}_{\sigma_i} - \frac{h}{2} \big( \boldsymbol{f}(\mathbf{x}_{\sigma_i}; \sigma_i) + \boldsymbol{f}(\mathbf{x}_{\sigma_{i+1}}^{\text{Euler}}; \sigma_{i+1}) \big), \tag{5}$$

$$\text{LTE}^{\text{Heun}} = \mathbf{x}_{\sigma_{i+1}} - \mathbf{x}_{\sigma_{i+1}}^{\text{Heun}} = \mathcal{O}(h^3). \tag{6}$$

This second-order Runge–Kutta method serves as the default solver throughout our work.

**Embedded Runge–Kutta pair.** In Heun's method, the Euler solution is computed first and then corrected. Thus, we obtain two solutions of different orders (Euler of order 1 and Heun of order 2). This structure is referred to as an *embedded Runge–Kutta pair*, and their solution difference $\Delta^{\mathbf{x}} := \mathbf{x}_{\sigma_{i+1}}^{\text{Heun}} - \mathbf{x}_{\sigma_{i+1}}^{\text{Euler}}$ is commonly used as a proxy for the LTE Hairer et al. (1993). In this paper, we refer to this difference $\Delta^{\mathbf{x}}$ as the **ERK solution difference**. We also define the **ERK drift difference** as the difference of the drift evaluated at the two solutions, $\Delta^{\mathbf{f}} := \boldsymbol{f}(\mathbf{x}_{\sigma_{i+1}}^{\text{Heun}}; \sigma_{i+1}) - \boldsymbol{f}(\mathbf{x}_{\sigma_{i+1}}^{\text{Euler}}; \sigma_{i+1})$.

**Stiffness.** Stiffness refers to the presence of both fast and slow dynamics within an ODE system Hairer & Wanner (1996). It is commonly encountered in physical simulations such as fluid dynamics. To ensure stability, numerical solvers must reduce their step sizes, leading to increased function evaluations and higher computational cost. To handle this, previous adaptive solvers Petzold (1983); Shampine & Gear (1979) detect stiffness and dynamically adjust their integration behavior by refining step sizes or switching to implicit solvers. Classically, stiffness is quantified by spectral properties of the Jacobian of the drift, such as the ratio of the largest and smallest eigenvalue magnitudes or, more simply, the maximum eigenvalue in magnitude Hairer & Wanner (1996):

$$J(\mathbf{x}_\sigma; \sigma) := \nabla_{\mathbf{x}_\sigma} \boldsymbol{f}(\mathbf{x}_\sigma; \sigma), \tag{7}$$

$$\rho_{\text{stiff}}(\mathbf{x}_\sigma, \sigma) := \max_k \left| \lambda_k\big(J(\mathbf{x}_\sigma, \sigma)\big) \right|, \tag{8}$$

where $\lambda_k(J)$ denotes the $k$-th eigenvalue of the matrix $J$. We denote by $\mathbf{v}_{\text{stiff}}(\mathbf{x}_\sigma, \sigma)$ a unit dominant eigenvector associated with $\rho_{\text{stiff}}(\mathbf{x}_\sigma, \sigma)$.

## 4 METHOD

We start with theoretical analysis and empirical evidence to provide the insight that, in stiff ODEs, both the local truncation error (LTE) and the embedded Runge–Kutta (ERK) solution difference are aligned with the Jacobian's dominant eigenvector (Section 4.1). Based on this observation, we introduce cost-free estimators for stiffness and the dominant eigenvector (Section 4.2), and incorporate them into our guidance scheme, ERK-Guid (Section 4.3).

### 4.1 ALIGNMENT OF LTE AND ERK SOLUTION DIFFERENCES IN STIFF ODES

**Theoretical Insight.** We assume that the score-based vector field of the diffusion model $\mathbf{x}_\sigma$ is well approximated by its local linearization around the current state $\mathbf{x}_{\sigma_i}$ when the step size $h := \sigma_i - \sigma_{i+1} > 0$ is sufficiently small:

$$\frac{\mathrm{d}\mathbf{x}_\sigma}{\mathrm{d}\sigma} = \boldsymbol{f}(\mathbf{x}_\sigma; \sigma) \approx \boldsymbol{f}(\mathbf{x}_{\sigma_i}; \sigma_i) + J(\mathbf{x}_{\sigma_i}; \sigma_i)(\mathbf{x}_\sigma - \mathbf{x}_{\sigma_i}). \tag{9}$$

Let $J_{\sigma_i}$ and $\boldsymbol{f}_{\sigma_i}$ denote $J(\mathbf{x}_{\sigma_i}; \sigma_i)$ and $\boldsymbol{f}(\mathbf{x}_{\sigma_i}; \sigma_i)$, respectively. From Eq. 1, the Jacobian is given by $J_{\sigma_i} = -\sigma_i \nabla^2_{\mathbf{x}_\sigma} \log p(\mathbf{x}_\sigma; \sigma)|_{\mathbf{x}_{\sigma_i}}$. Assuming $-\sigma \log p(\mathbf{x}_\sigma; \sigma)$ is $C^2$-smooth, its Hessian $J_{\sigma_i}$ is symmetric and thus admits the following eigendecomposition:

$$J_{\sigma_i} = V \Lambda V^\top, \quad \text{s.t.} \quad V^\top V = I, \quad J_{\sigma_i} \mathbf{v}_k = \lambda_k \mathbf{v}_k, \quad \forall k \tag{10}$$

where $V$ is the orthogonal matrix whose columns are the eigenvectors $\mathbf{v}_k$ of $J_{\sigma_i}$, $\Lambda$ is the diagonal matrix of the corresponding eigenvalues $\lambda_k$, and $I$ is the identity matrix. Therefore, the single-step Euler update can be decomposed along the eigenbasis as:

$$\mathbf{x}_{\sigma_{i+1}}^{\text{Euler}} - \mathbf{x}_{\sigma_i} = -h\boldsymbol{f}_{\sigma_i} = -h\sum_k \langle \boldsymbol{f}_{\sigma_i}, \mathbf{v}_k \rangle \mathbf{v}_k, \tag{11}$$

where $\langle \cdot, \cdot \rangle$ refers inner product. Similarly, Heun's update step is given by

$$\mathbf{x}_{\sigma_{i+1}}^{\text{Heun}} - \mathbf{x}_{\sigma_i} = -\frac{h}{2}\big(\boldsymbol{f}_{\sigma_i} + \boldsymbol{f}_{\mathbf{x}_{\sigma_{i+1}}}^{\text{Euler}}\big) \approx -\frac{h}{2}\big(\boldsymbol{f}_{\sigma_i} + \boldsymbol{f}_{\sigma_i} + J_{\sigma_i}(\mathbf{x}_{\sigma_{i+1}}^{\text{Euler}} - \mathbf{x}_{\sigma_i})\big) \tag{12}$$

$$= -h\sum_k \left(1 + \tfrac{1}{2}z_k\right) \langle \boldsymbol{f}_{\sigma_i}, \mathbf{v}_k \rangle \mathbf{v}_k, \tag{13}$$

where $\boldsymbol{f}_{\sigma_{i+1}}^{\text{Euler}} = \boldsymbol{f}(\mathbf{x}_{\sigma_{i+1}}^{\text{Euler}}; \sigma_{i+1})$ and $z_k = -h\lambda_k$. The exact update can be approximated as follows:

$$\mathbf{x}_{\sigma_{i+1}} - \mathbf{x}_{\sigma_i} \approx -h\sum_k \phi(z_k) \langle \boldsymbol{f}_{\sigma_i}, \mathbf{v}_k \rangle \mathbf{v}_k, \tag{14}$$

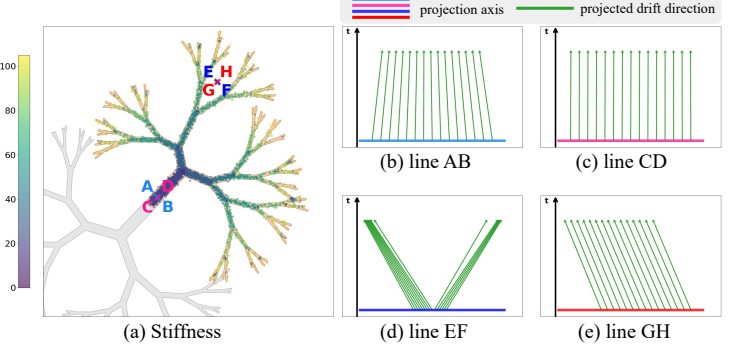

Table 1: **Projection of local truncation error (LTE) and ERK solution difference onto eigenvector axes.**

|  | LTE | ERK solution diff |
|---|---|---|
| EF | **7.22e-05** | **4.03e-04** |
| GH | 5.78e-06 | 6.18e-05 |
| Ratio | **12.5×** | **6.5×** |

Figure 1: **Toy 2D example of stiffness and local drift variations.** **(a)** Visualization of stiffness across the 2D plane, colored by magnitude. We sample a non-stiff region (lines AB and CD) and a stiff region (lines EF and GH). **(b, c)** In the non-stiff region, the drift projected onto the local eigenbasis exhibits minimal variation along both axes, indicating stable dynamics. **(d, e)** Conversely, in the stiff region, the projected drift shows pronounced variation along the *dominant* eigenvector axis (d), while remaining relatively parallel along the *subdominant* axis (e). **Table 1** shows that for a single step in the stiff region, both the local truncation error (LTE) and the ERK solution difference are heavily amplified along the dominant axis (EF) due to these drift variations. This strong alignment validates the ERK difference as a practical proxy for the severe LTE in our proposed ERK-Guid.

where the function $\phi(z) = \frac{e^z - 1}{z}$ for $z \neq 0$ and $\phi(0) = 1$. See Appendix A.1 for the derivation.

Thus, by subtracting Eq. 13 from Eq. 14 and Eq. 11 from Eq. 13, respectively, the local truncation error of Heun's method and the ERK solution difference can be written as:

$$\text{LTE}^{\text{Heun}} := \mathbf{x}_{\sigma_{i+1}} - \mathbf{x}_{\sigma_{i+1}}^{\text{Heun}} \approx -h \sum_k \alpha(z_k) \langle \boldsymbol{f}_{\sigma_i}, \mathbf{v}_k \rangle \mathbf{v}_k, \tag{15}$$

$$\Delta^{\mathbf{x}} := \mathbf{x}_{\sigma_{i+1}}^{\text{Heun}} - \mathbf{x}_{\sigma_{i+1}}^{\text{Euler}} \approx -h \sum_k \left(\tfrac{1}{2} z_k\right) \langle \boldsymbol{f}_{\sigma_i}, \mathbf{v}_k \rangle \mathbf{v}_k, \tag{16}$$

where $\alpha(z) = \frac{e^z - 1}{z} - 1 - \frac{1}{2}z$ for $z \neq 0$ and $\alpha(0) = 0$. Figure 6a visualizes the behavior of $\alpha(z)$.

As $|z_k| = |h\lambda_k|$ increases, the weights $\frac{1}{2} z_k$ and $\alpha(z_k)$ associated with each eigenvector component also grow in magnitude, so that contributions from directions with large $|\lambda_k|$ come to dominate. Consequently, both the local truncation error (LTE) and the ERK solution difference tend to align with the dominant eigenvector corresponding to the largest eigenvalue magnitude, specifically in stiff regions, i.e., regions with high stiffness. Motivated by this observation, we estimate both stiffness and the dominant eigenvector from the ERK solution difference during sampling (Section 4.2), and when stiffness is sufficiently high, we apply guidance with Eq. 15 (Section 4.3).

**Toy 2D Experiments.** To better understand the connection between ODE dynamics and stiffness, we construct a two-dimensional toy system with an analytically defined ground-truth drift, followed by Autoguidance Karras et al. (2024a). Experimental details, including the computation of eigenvectors, eigenvalues, and LTE, are provided in Appendix B. Figure 1(a) visualizes the degree of stiffness across the 2D plane. To examine the local drift behavior, we sample a non-stiff region (lines AB and CD) and a stiff region (lines EF and GH). For each region, we compute the Jacobian at its center and project the drift field onto its eigenvectors (eigenbasis), as shown in panels (b)–(e). In the non-stiff region (panels (b) and (c)), the projected drift exhibits minimal variation along both axes, indicating locally stable dynamics that numerical solvers can easily approximate. Conversely, in the stiff region, the drift shows pronounced variations along the *dominant* eigenvector axis (panel (d), line EF), while remaining relatively parallel along the *subdominant* axis (panel (e), line GH). Consequently, this variation causes ODE solvers to incur significant local truncation errors (LTE) predominantly along the dominant eigenvector direction. To verify this, we evaluate a single-step LTE using the center of the stiff region as the initial point. As presented in Table 1, the LTE projected onto the dominant eigenvector direction (line EF, 7.22e-05) is approximately **12.5×** larger than that along the subdominant direction (line GH, 5.78e-06).

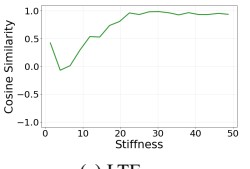 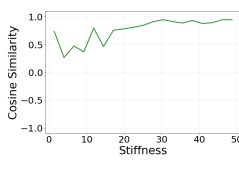 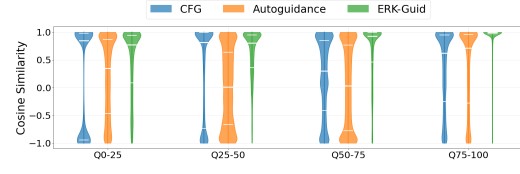

(a) LTE  (b) ERK solution difference  (c) ERK drift difference across stiffness quantiles

Figure 2: **Eigenvector alignment across stiffness.** **(a)** Cosine similarity between the dominant eigenvector and the local truncation error (LTE) increases with stiffness. **(b)** The ERK solution difference exhibits a similar trend to LTE, suggesting it can serve as a reliable proxy for the LTE direction in high stiffness regions. **(c)** Our ERK-Guid consistently achieves higher cosine similarity with the eigenvector, highlighting its strong alignment in stiff regions. CFG and Autoguidance exhibit weaker or mixed alignment with the dominant eigenvector in stiff regions, supporting the complementarity of our method.

To mitigate the severe LTE occurring predominantly along the dominant eigenvector, we need a practical proxy to estimate this error direction. For this, we introduce the ERK solution difference, which measures the gap between a higher-order and a lower-order solver starting from the same state. Because a higher-order solver applies much larger corrections along the dominant axis where the drift varies drastically, this difference vector naturally aligns with that axis. Table 1 confirms this intuition: the ERK solution difference along the dominant direction (line EF, 4.03e-04) is roughly **6.5×** larger than that along the subdominant direction (line GH, 6.18e-05). This motivates our core idea: utilizing the ERK difference to suppress severe errors in stiff regions.

For a broader analysis, we measure the cosine similarity of the dominant eigenvector with both the LTE and the ERK solution difference across different stiffness levels. As shown in Figure 2 (a), the alignment between the LTE and the dominant eigenvector steadily increases with stiffness. This demonstrates that the eigenvector serves as a reliable proxy for the LTE direction in stiff regions. Additionally, Figure 2 (b) illustrates that the ERK solution difference also strongly aligns with the dominant eigenvector when stiffness is high. This crucial insight motivates our novel guidance strategy, *ERK-Guid*, which leverages the dominant eigenvector estimated from the embedded Runge–Kutta (ERK) pair as its guiding signal.

We evaluate ERK-Guid against two widely used baselines: CFG Ho & Salimans (2022) (conditional–unconditional score difference) and Autoguidance Karras et al. (2024a) (main–weak model difference). Figure 2 (c) shows that ERK-Guid consistently maintains strong alignment with the dominant eigenvector across all stiffness quantiles. Notably, the performance gap widens in the high-stiffness bin (Q75–100), where CFG and Autoguidance exhibit weak or mixed alignment. These results suggest that ERK-Guid provides an orthogonal guidance signal that effectively complements, rather than overlaps with, existing methods.

### 4.2 STIFFNESS AND DOMINANT EIGENVECTOR ESTIMATOR

The key intuition from the previous section is that, once stiffness is high, the dominant eigenvector provides a reliable proxy for reducing local truncation error (LTE). In practice, however, direct access to the Jacobian is infeasible, making stiffness estimation challenging. A common alternative is to use Jacobian–vector product (JVP) based power iterations, which are supported in frameworks such as PyTorch but remain prohibitively expensive for diffusion sampling, where each step requires costly network evaluations. To overcome this, we propose cost-free estimators of stiffness and the dominant eigenvector, exploiting the ERK solution/drift difference without additional evaluations.

Let $\mathbf{x}_{\sigma_i}^{\text{Heun}}$ and $\mathbf{x}_{\sigma_i}^{\text{Euler}}$ denote the states at $\sigma_i$ computed by Heun's update and the intermediate Euler step from $\sigma_{i-1}$, respectively. We define the stiffness estimator as

$$\hat{\rho}_{\text{stiff}}\left(\mathbf{x}_{\sigma_i}^{\text{Heun}}, \mathbf{x}_{\sigma_i}^{\text{Euler}}, \sigma_i\right) := \frac{\left\| \boldsymbol{f}\left(\mathbf{x}_{\sigma_i}^{\text{Heun}}; \sigma_i\right) - \boldsymbol{f}\left(\mathbf{x}_{\sigma_i}^{\text{Euler}}; \sigma_i\right) \right\|_2}{\left\| \mathbf{x}_{\sigma_i}^{\text{Heun}} - \mathbf{x}_{\sigma_i}^{\text{Euler}} \right\|_2}. \tag{17}$$

Here, $\mathbf{x}_{\sigma_i}^{\text{Heun}} - \mathbf{x}_{\sigma_i}^{\text{Euler}}$ corresponds exactly to the ERK solution difference, while $\boldsymbol{f}\left(\mathbf{x}_{\sigma_i}^{\text{Heun}}; \sigma_i\right) - \boldsymbol{f}\left(\mathbf{x}_{\sigma_i}^{\text{Euler}}; \sigma_i\right)$ represents the ERK drift difference.

**Proposition 1** *Let $J_{\sigma_i}$ be the Jacobian matrix of the drift function $\boldsymbol{f}(\mathbf{x}_\sigma; \sigma)$ evaluated at $\mathbf{x}_{\sigma_i}^{\text{Heun}}$. Assume that $\boldsymbol{f}(\mathbf{x}_\sigma; \sigma)$ has a locally Lipschitz Jacobian near $\mathbf{x}_{\sigma_i}^{\text{Heun}}$. If the ERK solution difference $\Delta^{\mathbf{x}} := \mathbf{x}_{\sigma_i}^{\text{Heun}} - \mathbf{x}_{\sigma_i}^{\text{Euler}}$ is a nonzero vector, sufficiently small, and aligned with the eigenvector associated with the dominant eigenvalue $\lambda$ of $J_{\sigma_i}$ in magnitude, such that*

$$\|J_{\sigma_i}(\mathbf{x}_{\sigma_i}^{\text{Heun}} - \mathbf{x}_{\sigma_i}^{\text{Euler}})\|_2 = |\lambda|\|\mathbf{x}_{\sigma_i}^{\text{Heun}} - \mathbf{x}_{\sigma_i}^{\text{Euler}}\|_2 + \mathcal{O}(\|\Delta^{\mathbf{x}}\|_2^2),$$

*then the magnitude of the dominant eigenvalue, $|\lambda|$, admits the approximation*

$$|\lambda| = \frac{\left\|\boldsymbol{f}(\mathbf{x}_{\sigma_i}^{\text{Heun}}; \sigma_i) - \boldsymbol{f}(\mathbf{x}_{\sigma_i}^{\text{Euler}}; \sigma_i)\right\|_2}{\left\|\mathbf{x}_{\sigma_i}^{\text{Heun}} - \mathbf{x}_{\sigma_i}^{\text{Euler}}\right\|_2} + \mathcal{O}(\|\Delta^{\mathbf{x}}\|_2).$$

Proposition 1 establishes that the proposed stiffness estimator accurately recovers the true stiffness under the assumption. We provide the proof of Proposition 1 in Appendix A.2. Since our estimator relies on the alignment between the ERK solution difference and the dominant eigenvector, it is reliable only when the estimated stiffness is sufficiently high. This requirement is explicitly incorporated into our guidance design (see Section 4.3). Notably, the stiffness estimator requires no additional network evaluations: $\mathbf{x}_{\sigma_i}^{\text{Heun}}$, $\mathbf{x}_{\sigma_i}^{\text{Euler}}$, and $\boldsymbol{f}(\mathbf{x}_{\sigma_i}^{\text{Euler}}; \sigma_i)$ are already obtained during the Heun correction, while $\boldsymbol{f}(\mathbf{x}_{\sigma_i}^{\text{Heun}}; \sigma_i)$ is reused in the subsequent Huen step.

As shown in Section 4.1, the ERK solution difference provides a useful proxy for the dominant eigenvector in stiff regions. To improve robustness, we define the dominant eigenvector estimator as normalized *ERK drift difference*:

$$\hat{\mathbf{v}}_{\text{stiff}}(\mathbf{x}_{\sigma_i}^{\text{Heun}}, \mathbf{x}_{\sigma_i}^{\text{Euler}}, \sigma_i) := \frac{\boldsymbol{f}(\mathbf{x}_{\sigma_i}^{\text{Heun}}; \sigma_i) - \boldsymbol{f}(\mathbf{x}_{\sigma_i}^{\text{Euler}}; \sigma_i)}{\left\|\boldsymbol{f}(\mathbf{x}_{\sigma_i}^{\text{Heun}}; \sigma_i) - \boldsymbol{f}(\mathbf{x}_{\sigma_i}^{\text{Euler}}; \sigma_i)\right\|_2}. \tag{18}$$

Under a local linearization, this difference approximates the Jacobian applied to the ERK solution difference:

$$\boldsymbol{f}(\mathbf{x}_{\sigma_i}^{\text{Heun}}; \sigma_i) - \boldsymbol{f}(\mathbf{x}_{\sigma_i}^{\text{Euler}}; \sigma_i) \approx J_{\sigma_i}(\mathbf{x}_{\sigma_i}^{\text{Heun}}; \sigma_i)(\mathbf{x}_{\sigma_i}^{\text{Heun}} - \mathbf{x}_{\sigma_i}^{\text{Euler}}),$$

which corresponds to a single-step JVP power iteration. In the eigenbasis of $J_{\sigma_i}$, components with larger eigenvalues are amplified, effectively suppressing subdominant directions and naturally steering the estimate toward the dominant eigenvector. We demonstrate this in Section 5.1.

## 4.3 Embedded Runge–Kutta Guidance

We now refine Eq. 15 into a practical guidance scheme. Leveraging cost-free estimates of stiffness and the dominant eigenvector, we construct a stabilized variant that ensures consistent updates across stiffness levels. The modified formulation adaptively scales the correction according to the estimated stiffness and aligns the update with the dominant local dynamics.

Using the proposed estimators, we denote $\boldsymbol{f}_{\sigma_i}^{\text{Heun}} := \boldsymbol{f}(\mathbf{x}_{\sigma_i}^{\text{Heun}}; \sigma_i)$, $\hat{\rho}_{\sigma_i} := \hat{\rho}_{\text{stiff}}(\mathbf{x}_{\sigma_i}^{\text{Heun}}, \mathbf{x}_{\sigma_i}^{\text{Euler}}, \sigma_i)$, and $\hat{\mathbf{v}}_{\sigma_i} := \hat{\mathbf{v}}_{\text{stiff}}(\mathbf{x}_{\sigma_i}^{\text{Heun}}, \mathbf{x}_{\sigma_i}^{\text{Euler}}, \sigma_i)$. With step size $h := \sigma_i - \sigma_{i+1} > 0$, we define the ERK-Guid update as:

$$\hat{\mathbf{x}}_{\sigma_{i+1}}^{\text{Heun}} = \mathbf{x}_{\sigma_{i+1}}^{\text{Heun}} - h\,\beta\,z^2\left\langle \boldsymbol{f}_{\sigma_i}^{\text{Heun}}, \hat{\mathbf{v}}_{\sigma_i}\right\rangle \hat{\mathbf{v}}_{\sigma_i}, \tag{19}$$

where $\beta = \mathbf{1}_{\{\hat{\rho}_{\sigma_i} > w_{\text{con}}\}}$ is a binary indicator that activates the guidance when the estimated stiffness exceeds the prescribed threshold $w_{\text{con}}$, and $z := w_{\text{stiff}}\,h\,\hat{\rho}_{\sigma_i}$ adaptively scales the guidance magnitude according to the estimated stiffness. Here, $w_{\text{stiff}}$ is a hyperparameter that scales the overall guidance strength. Importantly, this update can be equivalently written in the form of conventional guidance:

$$\hat{\mathbf{x}}_{\sigma_{i+1}}^{\text{Heun}} = \mathbf{x}_{\sigma_{i+1}}^{\text{Heun}} - h\,\gamma\left(\boldsymbol{f}_{\sigma_i}^{\text{Heun}} - \boldsymbol{f}_{\sigma_i}^{\text{Euler}}\right), \tag{20}$$

where $\gamma = \frac{\beta\,z^2\left\langle \boldsymbol{f}_{\sigma_i}^{\text{Heun}}, \hat{\mathbf{v}}_{\sigma_i}\right\rangle}{\left\|\boldsymbol{f}_{\sigma_i}^{\text{Heun}} - \boldsymbol{f}_{\sigma_i}^{\text{Euler}}\right\|_2}$, and $\boldsymbol{f}_{\sigma_i}^{\text{Euler}} = \boldsymbol{f}(\mathbf{x}_{\sigma_i}^{\text{Euler}}; \sigma_i)$. This form reveals that ERK-Guid operates as a guidance term that extrapolates between two model predictions, analogous to conventional guidance mechanisms, while grounding the guidance direction in an estimator of the local truncation

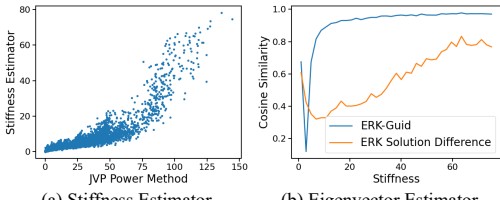

Figure 3: **Accuracy of proposed estimators.** **(a)** Our estimated stiffness values highly correlate with the JVP-based one. **(b)** ERK drift difference (blue) maintains higher alignment with the dominant eigenvector than the ERK solution difference (orange), especially at high stiffness.

Figure 4: **Grid search of hyperparameters.** Quantitative trends of FID and FD-DINOv2 as $w_{con}$ varies. Each curve corresponds to a different value of $w_{stiff}$, with points indicating increasing $w_{con}$, shown for (a) 16-step and (b) 32-step sampling.

Table 2: **Quantitative results on ImageNet-512.** Rows with $w_{stiff} = 0$ indicate no guidance, while nonzero $w_{stiff}$ correspond to ERK-Guid.

| #step | $w_{stiff}$ | FD-DINOv2 ↓ | FID ↓ | Precision ↑ | Recall ↑ | IS ↑ |
|---|---|---|---|---|---|---|
| 32 | 0.0 | 90.1 | 2.58 | 0.631 | 0.672 | 244 |
| 32 | 0.5 | 88.9 | 2.57 | 0.632 | 0.673 | 245 |
| 32 | 1.0 | 86.2 | **2.56** | **0.635** | 0.673 | 247 |
| 32 | 1.5 | 83.7 | 2.60 | **0.635** | **0.675** | 249 |
| 32 | 2.0 | **82.8** | 2.74 | 0.632 | 0.674 | 247 |
| 32 | 2.5 | 84.9 | 3.03 | 0.625 | 0.667 | 241 |
| 16 | 0.0 | 97.4 | 2.79 | 0.629 | 0.652 | 238 |
| 16 | 0.75 | **88.9** | **2.68** | **0.639** | **0.659** | **244** |
| 8 | 0.0 | 161.2 | 7.06 | 0.446 | **0.614** | 183 |
| 8 | 0.5 | **136.9** | **4.91** | **0.545** | 0.598 | **201** |

Table 3: **Quantitative results of guidance compatibility on ImageNet-512.** ERK-Guid is combined with established guidance methods to enhance sampling quality and robustness under various guidance configurations.

| #step | Method | FD-DINOv2 ↓ | FID ↓ | Precision ↑ | Recall ↑ | IS ↑ |
|---|---|---|---|---|---|---|
| 32 | CFG | 88.5 | **2.27** | 0.609 | **0.707** | 271 |
| | **+ERK-Guid** | **83.9** | **2.27** | **0.612** | **0.707** | **275** |
| 32 | Autoguidance | 50.4 | **1.36** | 0.691 | 0.628 | 262 |
| | **+ERK-Guid** | **47.6** | **1.36** | **0.694** | **0.630** | **268** |
| 16 | CFG | 133.9 | 3.61 | 0.593 | **0.673** | 210 |
| | **+ERK-Guid** | **125.6** | **3.20** | **0.606** | **0.673** | **215** |
| 16 | Autoguidance | 82.2 | 2.32 | 0.652 | 0.641 | 229 |
| | **+ERK-Guid** | **75.2** | **1.93** | **0.669** | **0.644** | **236** |

error. Derivations from Eq. 15 to Eq. 19 and from Eq. 19 to Eq. 20 are provided in Appendix A.3 and Appendix A.4, respectively.

Compared to the exact LTE expression, our final formulation incorporates a confidence gate $\beta$ to restrict guidance to sufficiently stiff regions, as analyzed in Figure 4. It also includes a stiffness-dependent scaling controlled by the hyperparameter $w_{stiff}$, which adjusts the overall correction strength, with no guidance applied when $w_{stiff} = 0$. In addition, we replace $\alpha(z)$ with a quadratic form $z^2$, which prevents excessive amplification under inaccurate estimates while preserving smooth behavior near zero. The choice of modulation is examined through ablations in Appendix B.3 and Figure 6b. The complete procedure is summarized in Algorithm 1.

**Computation cost.** ERK-Guid incurs no additional network evaluations: all required quantities are already computed during the Heun update. Thus, unlike CFG or Autoguidance, our approach imposes no extra evaluation overhead and relies solely on the discrepancy between two solver orders.

## 5 EXPERIMENTS

We evaluate ERK-Guid on both synthetic and real-world datasets. On synthetic data, we validate our stiffness estimator against Jacobian Vector Product (JVP) references and compare ERK solution and drift differences for eigenvector estimation in Section 5.1. In Section 5.2, we present quantitative results on real-world datasets, comparing our method against unguided sampling. In Section 5.3, we examine ERK-Guid's compatibility with existing guidance methods and its plug-and-play adaptability across different solvers to demonstrate its versatility. Furthermore, we compare ERK-Guid against classical stiffness-aware adaptive step-size control (Appendix B.6), which reduces the step size when high stiffness is detected, as well as predictor–corrector sampling (Appendix B.7), positioning our method against established numerical strategies. Detailed experimental settings and computational overhead analysis are provided in Appendix B.2.

Table 4: **Quantitative results of plug-and-play adaptation of ERK-Guid to solver methods on ImageNet-64 and FFHQ-64.**

| Dataset | ImageNet 64×64 (FID↓) | | | FFHQ 64×64 (FID↓) | | |
| NFEs | 6 | 8 | 10 | 6 | 8 | 10 |
| --- | --- | --- | --- | --- | --- | --- |
| Heun Hairer et al. (1993) | 89.63 | 37.65 | 16.46 | 142.4 | 57.21 | 29.54 |
| + **ERK-Guid** | **85.19** | **35.93** | **13.85** | **132.8** | **54.72** | **23.36** |
| DPM-Solver Lu et al. (2022) | 44.83 | 12.42 | 6.84 | 83.17 | 22.84 | 9.46 |
| + **ERK-Guid** | **31.59** | **10.57** | **6.54** | **49.0** | **10.42** | **4.64** |
| DEIS Zhang & Chen (2023) | 12.57 | 6.84 | 5.34 | 12.25 | 7.59 | 5.56 |
| + **ERK-Guid** | **9.56** | **6.25** | **4.89** | **9.96** | **6.04** | **4.46** |

## 5.1 ACCURACY OF THE ESTIMATORS

In Figure 3, we validate the accuracy of our estimators. Figure 3(a) illustrates that the stiffness estimator shows a strong correlation with the JVP reference, increasing consistently with the reference values. Also, Figure 3(b) demonstrates that the eigenvector estimator based on ERK drift differences exhibits higher alignment with the dominant eigenvector than the ERK solution difference, particularly in stiff regions. These results confirm the reliability of our estimators for identifying the dominant eigenvector direction as guidance.

## 5.2 EFFECTIVENESS OF THE GUIDANCE

Table 2 summarizes quantitative results on ImageNet $512{\times}512$ with EDM2 and the Heun sampler. We take $w_{\text{stiff}} = 0$ as the baseline without guidance. As the guidance scale increases, FD-DINOv2 consistently decreases and reaches its best at $w_{\text{stiff}} = 2.0$, yielding 82.8 compared to the baseline 90.1. Importantly, this fidelity gain is achieved while keeping FID competitive and consistently improving Precision, Recall, and Inception Score, indicating that our update strengthens fidelity without sacrificing diversity or alignment. The advantage becomes more pronounced under fewer sampling steps, where local truncation errors dominate. With 16 steps, FD-DINOv2 improves from 97.4 to 88.9 and FID from 2.79 to 2.68, accompanied by gains in Precision and Inception Score. At 8 steps, the effect is even stronger: FD-DINOv2 drops from 161.2 to 136.9, FID from 7.06 to 4.91, with substantial boosts in Precision, Recall, and Inception Score. Overall, these results demonstrate that ERK-Guid effectively mitigates error accumulation in stiff regions, delivering consistent improvements across settings and providing particular advantages in low-step regimes, all without any additional training or model evaluations. Moreover, Figure 4 provides a grid-search analysis over $w_{\text{stiff}}$ and threshold $w_{\text{con}}$, illustrating robust trends across hyperparameter choices and confirming that ERK-Guid maintains stable improvements over a wide range of settings. Qualitative results across varying $w_{\text{stiff}}$ and $w_{\text{con}}$ are provided in Appendix E.

## 5.3 GUIDANCE COMPATIBILITY AND PLUG-AND-PLAY SOLVER ADAPTATION

In this section, we highlight two key properties of ERK-Guid: (i) its compatibility with existing model-based guidance methods, and (ii) its plug-and-play adaptability to various solvers.

**Guidance compatibility.** We examine whether ERK-Guid can be combined with existing guidance schemes. Diffusion sampling errors arise from two sources: model error and solver error (LTE). Under the predictor–corrector view Bradley & Nakkiran (2024), guidance methods act as correctors. ERK-Guid targets local truncation error (LTE), whereas CFG and Autoguidance primarily address model error arising from imperfect score estimates. Motivated by this complementarity, we combine ERK-Guid with CFG and Autoguidance in Table 3, showing that our correction consistently strengthens model-based guidance and generalizes well to other guidance methods. Additional details are provided in Appendix B.2.

**Plug-and-play adaptation.** Similar to other guidance methods, ERK-Guid can be applied to various solvers as a plug-and-play correction module. To demonstrate its effectiveness and broad applicability, we evaluate ERK-Guid on higher-order solvers, including Heun's method Hairer et al. (1993), DPM-Solver Lu et al. (2022), and DEIS Zhang & Chen (2023) on ImageNet Deng et al. (2009) and FFHQ Karras et al. (2019) at $64 \times 64$ resolution. Table 4 demonstrates that combining ERK-Guid with solver methods consistently improves performance across all NFEs on both datasets. These

DPM-Solver            DPM-Solver + ERK-Guid **(Ours)**

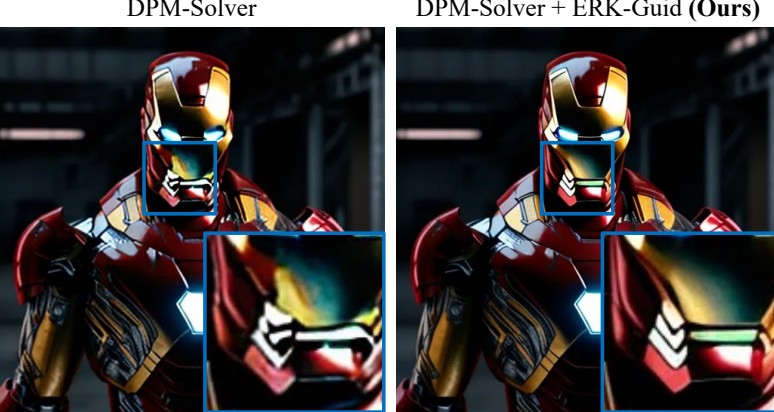

A hyper-realistic Iron Man suit with crisp metallic reflections and
glowing arc reactor in a dark hangar.

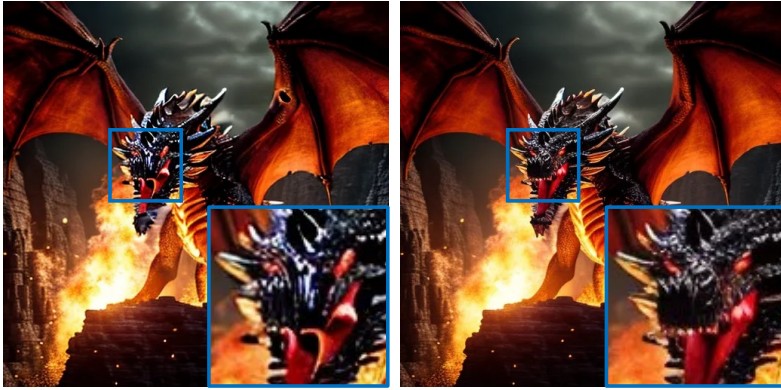

A massive dragon breathing a stream of bright fire in a dark canyon,
glowing scales and sharp wing edges, cinematic lighting.

Figure 5: **Qualitative comparison on PixArt-$\alpha$ Chen et al. (2023).** We perform text-to-image generation to compare DPM-Solver with our ERK-Guid. As shown in the blue zoomed-in regions, ERK-Guid captures fine semantic details more accurately.

results highlight the robust plug-and-play capability of ERK-Guid and its effectiveness even when paired with higher-order ODE solvers. Additional details are provided in Appendix B.5 and Table 9. Moreover, Figure 5 demonstrates that ERK-Guid delivers strong qualitative improvements on PixArt-$\alpha$ Chen et al. (2023), built upon a Diffusion Transformer (DiT) Peebles & Xie (2023) backbone, further highlighting its architectural generalization capability.

## 6    CONCLUSION

In this work, we proposed ERK-Guid, a stiffness-aware diffusion sampling framework that exploits Embedded Runge-Kutta solver discrepancies as informative guidance signals. Motivated by the alignment between local truncation errors and dominant eigenvectors in stiff regions, we introduce cost-free estimators for stiffness detection and eigenvector estimation based on ERK solution and drift differences, without additional network evaluations. Building on these estimators, we design a stabilized guidance scheme that provides an orthogonal correction and integrates seamlessly as a plug-and-play module into model-error-based guidance schemes and Runge-Kutta-based solvers without modifying their core updates. Extensive experiments demonstrate that ERK-Guid consistently improves sample quality and efficiency over strong baselines, and further show favorable comparisons against adaptive step-size control and predictor-corrector samplers. Overall, ERK-Guid establishes a principled and practical framework for stiffness-aware guidance in diffusion models, opening new directions that bridge numerical analysis and generative modeling.

REPRODUCIBILITY STATEMENT

In Section 4.3, we describe our pipeline design, and the Appendix C provides algorithmic details and full pseudocode. Appendix B documents experimental settings and hyperparameters. We provide the official implementation at `https://github.com/mlvlab/ERK-Guid`.

ETHICS STATEMENT

Our method operates at the sampling stage of pretrained diffusion models and does not introduce new training data or modify model parameters. As such, it inherits the potential risks of the underlying models, including biased, harmful, or inappropriate outputs depending on the conditioning input. Users and practitioners should ensure appropriate monitoring and alignment measures when applying ERK-Guid.

ACKNOWLEDGEMENT

This research was supported by the ASTRA Project through the National Research Foundation (NRF) funded by the Ministry of Science and ICT (No. RS-2024-00439619, 50%) and (No. RS-2024-00440063, 50%).

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

# A    DERIVATION

## A.1    DERIVATION OF THE EXACT ONE STEP INCREMENT

We provide a detailed derivation of Eq. 14, i.e., the exact one-step increment under the local linearization. Recall that around $\mathbf{x}_{\sigma_i}$, the score-based vector field is approximated as

$$\frac{\mathrm{d}\mathbf{x}_\sigma}{\mathrm{d}\sigma} \approx \boldsymbol{f}_{\sigma_i} + J_{\sigma_i}(\mathbf{x}_\sigma - \mathbf{x}_{\sigma_i}), \tag{21}$$

where we denote $J_{\sigma_i} = J(\mathbf{x}_{\sigma_i}; \sigma_i)$ and $\boldsymbol{f}_{\sigma_i} = \boldsymbol{f}(\mathbf{x}_{\sigma_i}; \sigma_i)$. This yields a linear ODE of the form

$$\frac{\mathrm{d}\mathbf{y}}{\mathrm{d}\sigma} = J_{\sigma_i}\mathbf{y} + \boldsymbol{f}_{\sigma_i}, \quad \mathbf{y}(\sigma_i) = \mathbf{0}, \tag{22}$$

where we introduced the shifted variable $\mathbf{y}(\sigma) = \mathbf{x}_\sigma - \mathbf{x}_{\sigma_i}$. Let $h := \sigma_i - \sigma_{i+1} > 0$ denote the step size. Our goal is to compute $\mathbf{y}(\sigma_i - h) = \mathbf{x}_{\sigma_{i+1}} - \mathbf{x}_{\sigma_i}$.

Assuming $-\sigma \log p(\mathbf{x}_\sigma; \sigma)$ is $C^2$-smooth, its Hessian $J_{\sigma_i}$ is symmetric and thus admits the following eigendecomposition:

$$J_{\sigma_i} = V\Lambda V^\top, \quad \text{s.t.} \quad V^\top V = I, \quad J_{\sigma_i}\mathbf{v}_k = \lambda_k\mathbf{v}_k, \quad \forall k \tag{23}$$

where $V$ is the orthogonal matrix whose columns are the eigenvectors $\mathbf{v}_k$ of $J_{\sigma_i}$, $\Lambda$ is the diagonal matrix of the corresponding eigenvalues $\lambda_k$, and $I$ is the identity matrix.

Let us define the state projected onto the eigenbasis space:

$$\boldsymbol{u}(\sigma) := V^\top\boldsymbol{y}(\sigma), \qquad \boldsymbol{g} := V^\top\boldsymbol{f}_{\sigma_i}, \qquad g_k = \boldsymbol{g}^\top\boldsymbol{e}_k = \langle \boldsymbol{f}_{\sigma_i}, \mathbf{v}_k \rangle, \tag{24}$$

where $\langle \cdot, \cdot \rangle$ refers to inner product, and $\boldsymbol{e}_k$ is the $k$-th standard basis vector.

Since $V$ is constant on this interval (by the local linearization assumption):

$$\frac{\mathrm{d}\boldsymbol{u}}{\mathrm{d}\sigma} = \frac{\mathrm{d}}{\mathrm{d}\sigma}\big(V^\top\boldsymbol{y}(\sigma)\big) = V^\top\frac{\mathrm{d}\boldsymbol{y}}{\mathrm{d}\sigma} \tag{25}$$

$$= V^\top(J_{\sigma_i}\boldsymbol{y} + \boldsymbol{f}_{\sigma_i}) \tag{26}$$

$$= V^\top(V\Lambda V^\top)\boldsymbol{y} + V^\top\boldsymbol{f}_{\sigma_i} \tag{27}$$

$$= (V^\top V)\Lambda(V^\top\boldsymbol{y}) + \boldsymbol{g} \tag{28}$$

$$= \Lambda\boldsymbol{u} + \boldsymbol{g}. \tag{29}$$

Therefore, the system *decouples* into independent scalar ODEs because $\Lambda$ is diagonal.

Writing the $k$-th coordinate explicitly,

$$\frac{\mathrm{d}u_k}{\mathrm{d}\sigma} = \lambda_k u_k + g_k, \qquad u_k(\sigma_i) = 0. \tag{30}$$

It has the closed-form solution:

$$u_k(\sigma) = (\sigma - \sigma_i)\,\phi\big(\lambda_k(\sigma - \sigma_i)\big)\,g_k, \tag{31}$$

where the function $\phi(z) = \frac{e^z - 1}{z}$ for $z \neq 0$ and $\phi(0) = 1$.

Let $z_k := \lambda_k(\sigma_{i+1} - \sigma_i) = -h\lambda_k$. Since $\mathbf{y} = V\mathbf{u}$, we obtain

$$\mathbf{x}_{\sigma_{i+1}} - \mathbf{x}_{\sigma_i} = \mathbf{y}(\sigma_i - h) = V\mathbf{u}(\sigma_i - h) \tag{32}$$

$$= -hV\sum_k \phi(z_k)\,g_k\mathbf{e}_k \tag{33}$$

$$= -h\sum_k \phi(z_k)\,g_k V\mathbf{e}_k \tag{34}$$

$$= -h\sum_k \phi(z_k)\,\langle\boldsymbol{f}_{\sigma_i}, \mathbf{v}_k\rangle\,\mathbf{v}_k, \tag{35}$$

which is the desired expression in Eq. 14.

## A.2 DERIVATION OF PROPOSITION 1

In Section 4.2 of the main paper, we introduce a cost-free stiffness estimator that exploits the ERK difference without additional evaluations. We provide the proof of Proposition 1, which shows that the proposed estimator can approximate the magnitude of the dominant eigenvalue.

**Proposition 1** *Let $J_{\sigma_i}$ be the Jacobian matrix of the drift function $\boldsymbol{f}(\mathbf{x}_\sigma; \sigma)$ evaluated at $\mathbf{x}_{\sigma_i}^{\text{Heun}}$. Assume that $\boldsymbol{f}(\mathbf{x}_\sigma; \sigma)$ has a locally Lipschitz Jacobian near $\mathbf{x}_{\sigma_i}^{\text{Heun}}$. If the ERK solution difference $\Delta^{\mathbf{x}} := \mathbf{x}_{\sigma_i}^{\text{Heun}} - \mathbf{x}_{\sigma_i}^{\text{Euler}}$ is a nonzero vector, sufficiently small, and aligned with the eigenvector associated with the dominant eigenvalue $\lambda$ of $J_{\sigma_i}$ in magnitude, such that*

$$\|J_{\sigma_i}(\mathbf{x}_{\sigma_i}^{\text{Heun}} - \mathbf{x}_{\sigma_i}^{\text{Euler}})\|_2 = |\lambda|\|\mathbf{x}_{\sigma_i}^{\text{Heun}} - \mathbf{x}_{\sigma_i}^{\text{Euler}}\|_2 + \mathcal{O}(\|\Delta^{\mathbf{x}}\|_2^2),$$

*then the magnitude of the dominant eigenvalue, $|\lambda|$, admits the approximation*

$$|\lambda| = \frac{\|\boldsymbol{f}(\mathbf{x}_{\sigma_i}^{\text{Heun}}; \sigma_i) - \boldsymbol{f}(\mathbf{x}_{\sigma_i}^{\text{Euler}}; \sigma_i)\|_2}{\|\mathbf{x}_{\sigma_i}^{\text{Heun}} - \mathbf{x}_{\sigma_i}^{\text{Euler}}\|_2} + \mathcal{O}(\|\Delta^{\mathbf{x}}\|_2).$$

**Proof 1** *Under the assumptions of the proposition, a first-order Taylor expansion of $\boldsymbol{f}(\mathbf{x}_\sigma; \sigma)$ about $\mathbf{x}_{\sigma_i}^{\text{Heun}}$ gives*

$$\boldsymbol{f}\big(\mathbf{x}_{\sigma_i}^{\text{Euler}}; \sigma_i\big) = \boldsymbol{f}\big(\mathbf{x}_{\sigma_i}^{\text{Heun}}; \sigma_i\big) - J_{\sigma_i}(\mathbf{x}_{\sigma_i}^{\text{Heun}} - \mathbf{x}_{\sigma_i}^{\text{Euler}}) + \mathbf{e}, \tag{36}$$

*where the remainder $\mathbf{e} = \mathcal{O}\big(\|\Delta^{\mathbf{x}}\|_2^2\big)$ is bounded due to the locally Lipschitz Jacobian.*

*Rearranging Eq. 36 to isolate the remainder $\mathbf{e}$ yields*

$$J_{\sigma_i}(\mathbf{x}_{\sigma_i}^{\text{Heun}} - \mathbf{x}_{\sigma_i}^{\text{Euler}}) - \big(\boldsymbol{f}(\mathbf{x}_{\sigma_i}^{\text{Heun}}; \sigma_i) - \boldsymbol{f}(\mathbf{x}_{\sigma_i}^{\text{Euler}}; \sigma_i)\big) = \mathbf{e}. \tag{37}$$

*Taking the vector norm on both sides gives*

$$\big\|J_{\sigma_i}(\mathbf{x}_{\sigma_i}^{\text{Heun}} - \mathbf{x}_{\sigma_i}^{\text{Euler}}) - \big(\boldsymbol{f}(\mathbf{x}_{\sigma_i}^{\text{Heun}}; \sigma_i) - \boldsymbol{f}(\mathbf{x}_{\sigma_i}^{\text{Euler}}; \sigma_i)\big)\big\|_2 = \|\mathbf{e}\|_2. \tag{38}$$

*Applying the reverse triangle inequality $\big|\|\mathbf{a}\|_2 - \|\mathbf{b}\|_2\big| \le \|\mathbf{a} - \mathbf{b}\|_2$ to Eq. 38, we obtain*

$$\left|\|J_{\sigma_i}(\mathbf{x}_{\sigma_i}^{\text{Heun}} - \mathbf{x}_{\sigma_i}^{\text{Euler}})\|_2 - \big\|\boldsymbol{f}(\mathbf{x}_{\sigma_i}^{\text{Heun}}; \sigma_i) - \boldsymbol{f}(\mathbf{x}_{\sigma_i}^{\text{Euler}}; \sigma_i)\big\|_2\right| \le \|\mathbf{e}\|_2 = \mathcal{O}\big(\|\Delta^{\mathbf{x}}\|_2^2\big), \tag{39}$$

*which directly implies*

$$\|J_{\sigma_i}(\mathbf{x}_{\sigma_i}^{\text{Heun}} - \mathbf{x}_{\sigma_i}^{\text{Euler}})\|_2 = \big\|\boldsymbol{f}(\mathbf{x}_{\sigma_i}^{\text{Heun}}; \sigma_i) - \boldsymbol{f}(\mathbf{x}_{\sigma_i}^{\text{Euler}}; \sigma_i)\big\|_2 + \mathcal{O}\big(\|\Delta^{\mathbf{x}}\|_2^2\big). \tag{40}$$

*Applying the alignment assumption to Eq. 40 yields*

$$|\lambda| \|\mathbf{x}_{\sigma_i}^{\text{Heun}} - \mathbf{x}_{\sigma_i}^{\text{Euler}}\|_2 = \big\|\boldsymbol{f}(\mathbf{x}_{\sigma_i}^{\text{Heun}}; \sigma_i) - \boldsymbol{f}(\mathbf{x}_{\sigma_i}^{\text{Euler}}; \sigma_i)\big\|_2 + \mathcal{O}\big(\|\Delta^{\mathbf{x}}\|_2^2\big). \tag{41}$$

*Since $\|\mathbf{x}_{\sigma_i}^{\text{Heun}} - \mathbf{x}_{\sigma_i}^{\text{Euler}}\|_2 > 0$, dividing both sides by this norm produces the desired approximation*

$$|\lambda| = \frac{\big\|\boldsymbol{f}(\mathbf{x}_{\sigma_i}^{\text{Heun}}; \sigma_i) - \boldsymbol{f}(\mathbf{x}_{\sigma_i}^{\text{Euler}}; \sigma_i)\big\|_2}{\|\mathbf{x}_{\sigma_i}^{\text{Heun}} - \mathbf{x}_{\sigma_i}^{\text{Euler}}\|_2} + \mathcal{O}\big(\|\Delta^{\mathbf{x}}\|_2\big). \tag{42}$$

## A.3 DERIVATION OF THE ERK-GUID FROM EQ. 15

We present a detailed derivation showing how the ERK-Guid (Eq. 19) can be obtained from the local truncation error of Heun's method (Eq. 15). Let $\lambda_1$ and $\mathbf{v}_1$ denote the dominant eigenvalue and its corresponding eigenvector of the Jacobian $J_{\sigma_i}$, respectively. As discussed in Section 4.1, when the dominant eigenvector $\mathbf{v}_1$ governs local dynamics, the LTE of Heun's method is dominated along the direction of $\mathbf{v}_1$:

$$\text{LTE}^{\text{Heun}} := \mathbf{x}_{\sigma_{i+1}} - \mathbf{x}_{\sigma_{i+1}}^{\text{Heun}} \approx -h\,\alpha(z_1)\langle \boldsymbol{f}_{\sigma_i}, \mathbf{v}_1\rangle\,\mathbf{v}_1, \tag{43}$$

where $z_1 := -h\lambda_1$. Rearranging terms gives

$$\mathbf{x}_{\sigma_{i+1}} \approx \mathbf{x}_{\sigma_{i+1}}^{\text{Heun}} - h\,\alpha(z_1)\langle \boldsymbol{f}_{\sigma_i}, \mathbf{v}_1\rangle\,\mathbf{v}_1. \tag{44}$$

The Taylor expansion of $\alpha(z)$ at $z = 0$ yields

$$\alpha(z_1) = \frac{1}{6}z_1^2 + \mathcal{O}(z_1^3), \tag{45}$$

and substituting this Taylor approximation into Eq. 44 gives

$$\mathbf{x}_{\sigma_{i+1}} \approx \mathbf{x}_{\sigma_{i+1}}^{\text{Heun}} - \frac{1}{6}h\,z_1^2\,\langle \boldsymbol{f}_{\sigma_i},\, \mathbf{v}_1 \rangle\,\mathbf{v}_1. \tag{46}$$

We interpret this additive term as a guidance correction and introduce a tunable scale $w_{\text{stiff}}$ as follows

$$\hat{\mathbf{x}}_{\sigma_{i+1}}^{\text{Heun}} = \mathbf{x}_{\sigma_{i+1}}^{\text{Heun}} - h\,z^2\,\langle \boldsymbol{f}_{\sigma_i},\, \mathbf{v}_1 \rangle\,\mathbf{v}_1, \quad z := w_{\text{stiff}}\,h\lambda_1, \tag{47}$$

where constant $\frac{1}{6}$ is absorbed into $w_{\text{stiff}}$.

In practice, as the sampling proceeds sequentially, the initial point for the current step is from the previous step's Heun update. We denote the drift evaluated at this initial point as $\boldsymbol{f}_{\sigma_i}^{\text{Heun}}$. Furthermore, since neither the exact dominant eigenvector $\mathbf{v}_1$ nor the eigenvalue $\lambda_1$ is available, we replace them with our cost-free estimators $\hat{\mathbf{v}}_{\sigma_i}$ and $\hat{\rho}_{\sigma_i}$ evaluated at this state. This yields the practical ERK-Guid update:

$$\hat{\mathbf{x}}_{\sigma_{i+1}}^{\text{Heun}} = \mathbf{x}_{\sigma_{i+1}}^{\text{Heun}} - h\,z^2\,\langle \boldsymbol{f}_{\sigma_i}^{\text{Heun}},\, \hat{\mathbf{v}}_{\sigma_i} \rangle\,\hat{\mathbf{v}}_{\sigma_i}, \quad z := w_{\text{stiff}}\,h\hat{\rho}_{\sigma_i}. \tag{48}$$

Since the derivation assumes operation within stiff regions, we introduce a binary indicator $\beta$ that activates the guidance only when the estimated stiffness exceeds a threshold, i.e., $\hat{\rho}_{\sigma_i} > w_{\text{con}}$.

$$\hat{\mathbf{x}}_{\sigma_{i+1}}^{\text{Heun}} = \mathbf{x}_{\sigma_{i+1}}^{\text{Heun}} - h\,\beta\,z^2\,\langle \boldsymbol{f}_{\sigma_i}^{\text{Heun}},\, \hat{\mathbf{v}}_{\sigma_i} \rangle\,\hat{\mathbf{v}}_{\sigma_i}, \quad z := w_{\text{stiff}}\,h\hat{\rho}_{\sigma_i}. \tag{49}$$

### A.4 RESEMBLANCE BETWEEN OUR ERK-GUID AND OTHER GUIDANCES

We provide the alternative (but equivalent) formulation of our proposed method as a common form of guidance schemes. In Eq. 19, the ERK-Guid updates the Heun prediction as

$$\hat{\mathbf{x}}_{\sigma_{i+1}}^{\text{Heun}} = \mathbf{x}_{\sigma_{i+1}}^{\text{Heun}} - h\,\beta\,z^2\,\langle \boldsymbol{f}_{\sigma_i}^{\text{Heun}},\, \hat{\mathbf{v}}_{\sigma_i} \rangle\,\hat{\mathbf{v}}_{\sigma_i}, \tag{50}$$

where $h = \sigma_i - \sigma_{i+1}$, $\beta = \mathbb{1}_{\{\hat{\rho}_{\sigma_i} > w_{\text{con}}\}}$, $z = w_{\text{stiff}}\,h\,\hat{\rho}_{\sigma_i}$. Substituting the definition of the eigenvector estimator $\hat{\mathbf{v}}_{\sigma_i}$ (Eq. 18) into the above ERK-Guid update yields

$$\hat{\mathbf{x}}_{\sigma_{i+1}}^{\text{Heun}} = \mathbf{x}_{\sigma_{i+1}}^{\text{Heun}} - \frac{h\,\beta\,z^2\,\langle \boldsymbol{f}_{\sigma_i}^{\text{Heun}},\, \hat{\mathbf{v}}_{\sigma_i} \rangle}{\left\| \boldsymbol{f}_{\sigma_i}^{\text{Heun}} - \boldsymbol{f}_{\sigma_i}^{\text{Euler}} \right\|_2}\left( \boldsymbol{f}_{\sigma_i}^{\text{Heun}} - \boldsymbol{f}_{\sigma_i}^{\text{Euler}} \right). \tag{51}$$

By grouping the scalar coefficients into an adaptive scaling factor

$$\gamma(\mathbf{x}_{\sigma_i}^{\text{Heun}}, \mathbf{x}_{\sigma_i}^{\text{Euler}}, \sigma_i) = \frac{\beta\,z^2\,\langle \boldsymbol{f}_{\sigma_i}^{\text{Heun}},\, \hat{\mathbf{v}}_{\sigma_i} \rangle}{\left\| \boldsymbol{f}_{\sigma_i}^{\text{Heun}} - \boldsymbol{f}_{\sigma_i}^{\text{Euler}} \right\|_2}, \tag{52}$$

the ERK-Guid update can be rewritten as

$$\hat{\mathbf{x}}_{\sigma_{i+1}}^{\text{Heun}} = \mathbf{x}_{\sigma_{i+1}}^{\text{Heun}} - h\,\gamma(\mathbf{x}_{\sigma_i}^{\text{Heun}}, \mathbf{x}_{\sigma_i}^{\text{Euler}}, \sigma_i)\left( \boldsymbol{f}_{\sigma_i}^{\text{Heun}} - \boldsymbol{f}_{\sigma_i}^{\text{Euler}} \right). \tag{53}$$

The above equation reveals that our method admits an interpretation as a guidance term with an adaptive scaling factor $\gamma$.

## B EXPERIMENTAL DETAILS

### B.1 2D TOY EXPERIMENT

In Section 4.1 of the main paper, we analyze a synthetic 2D distribution with analytically known scores to study how stiffness influences drift geometry and local truncation error. Figure 1 illustrates

drift behavior in non-stiff and stiff regions, Table 1 presents the norms of the LTE and ERK solution difference when projected onto the dominant and subdominant eigenvector axes, and Figure 2 presents quantitative alignment results.

**2D Visualization.** The target distribution $p_{\text{data}}(\mathbf{x}|c)$ forms a tree-like structure in the 2D plane, with probability density concentrated along the branch centerlines. Figure 1 (a) visualizes the degree of stiffness across the 2D plane at the 29th sampling step (out of 32). To explicitly compare the local dynamics between non-stiff and stiff regions, we plot the drift field projected onto the local eigenbasis for both the non-stiff region (lines AB and CD) and the stiff region (lines EF and GH).

In the non-stiff region on the main trunk (panels (b) and (c), lines AB and CD), the density varies smoothly. Thus, the projected drift exhibits minimal variation and consistently points toward the high-density interior, forming nearly parallel patterns. Conversely, the stiff region is centered on the low-density gap between two diverging branches. Panel (d) (line EF) traverses this gap between two sub-branches. Because points E and F are located close to distinct high-density branches, the vectors in the gap split into two groups, pointing toward either E or F. Panel (e) (line GH) aligns with the branch leading toward the central junction. Since point G is closer to the high-density core branch than point H, the vectors flow uniformly from H toward G. Note that lines EF and GH correspond to the dominant and subdominant eigenvector axes, respectively; this drift behavior naturally illustrates why drift variations are pronounced along the dominant axis, where the larger associated eigenvalue reflects higher sensitivity in the vector field across the density gap.

For Table 1, we evaluate a single-step local truncation error (LTE) and the ERK solution difference using the center of the stiff region as the initial point. The LTE is evaluated by comparing a single-step Heun solution with an approximate ground-truth solution obtained by subdividing the step into 100 finer substeps. The ERK solution difference is computed by comparing the Heun and Euler methods starting from the same initial point. Both the LTE and the ERK solution difference vectors are then projected onto the eigenvectors of the Jacobian evaluated at the starting point, effectively decomposing them along the dominant (EF) and subdominant (GH) axes.

**Eigenvector Alignment.** We explicitly construct the Jacobian by applying Jacobian–vector products (JVPs) to one-hot basis vectors, and compute its eigenvalues and eigenvectors directly. Since eigenvectors are sign-ambiguous, we orient them to point in the same half-space as the drift at each state, following Eq. 15. Sampling is performed without any additional guidance and always with the ground-truth score function. Because the exact ground-truth solution is not available, we approximate it by subdividing each original step into 100 smaller substeps with a much finer step size. The local truncation error (LTE) is defined relative to the Heun method under this reference. Figures 2 (a) and (b) are obtained by partitioning into stiffness bins and plotting the median cosine similarity within each bin.

## B.2 MAIN EXPERIMENTS

**Experimental setup.** We conduct experiments on ImageNet (ILSVRC2012) Deng et al. (2009) at resolutions $512\times512$ and $64\times64$, as well as on the FFHQ Karras et al. (2019) at $64\times64$. We use the pre-trained EDM Karras et al. (2022) and EDM2 Karras et al. (2024a) models. Heun's method is used as the base solver, and other solvers are incorporated through our plug-and-play module. Evaluation metrics follow prior work: *fidelity* via FD-DINOv2 Stein et al. (2023), FID Heusel et al. (2017), and Precision Kynkäänniemi et al. (2019); *diversity* via Recall Kynkäänniemi et al. (2019); and *condition alignment* using Inception Score (IS) Salimans et al. (2016). For all quantitative evaluations, we generate 50k samples per setting.

**Implementation details.** We estimate the dominant eigenvector and the stiffness using JVP-based power iteration with a random initialization and 300 iterations per timestep. Figure 7 visualizes the per-timestep convergence and indicates that 300 iterations are sufficient. As in the toy setup, we fix the eigenvector orientation via Eq. 15 so that it points toward the local drift; Figure 3(b) reports the median cosine similarity under this convention. For Table 2, we use the confidence threshold $w_{\text{con}} = 0.5$. For Table 3, we use $w_{\text{con}} = 0.5$, and set $w_{\text{stiff}} = 1.0$ and $0.75$ for sampling with 32 and 16 steps, respectively. For Table 4, we use $w_{\text{con}} = 0.05$, and $w_{\text{stiff}}$ is specified in Table 5. In Figure 5, we conduct 15 sampling steps for text-to-image generation on PixArt-$\alpha$ Chen et al. (2023) which adopts Diffusion Transformer (DiT) Peebles & Xie (2023) architecture.

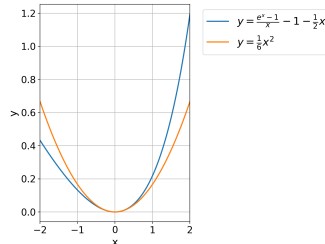 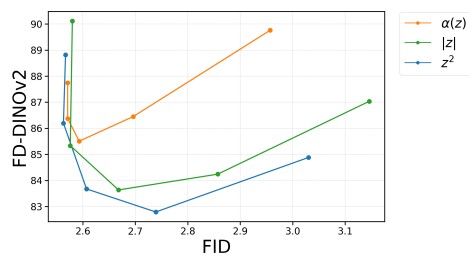

(a) **The behavior of the function** $y = \alpha(x)$ **and** $y = \frac{1}{6}x^2$.      (b) **Ablation on the scaling function of** $z$.

Figure 6: **Choice of ablation functions.** **(a)** Functional comparison of $\alpha(z)$ and its quadratic approximation. **(b)** Comparison of $\alpha(z)$, $|z|$, and $z^2$, under the FID–FD-DINOv2 trade-off, where the quadratic form $z^2$ achieves the most favorable balance between stability and fidelity.

**Computational cost.** All main experiments were run on $8\times$ NVIDIA RTX 3090 GPUs. In Table 6 and Table 7, we evaluate the wall-clock time and memory overhead on the ImageNet $512\times512$ dataset using a single RTX 3090 GPU with batch size 1, comparing ERK-Guid against Heun's method. ERK-Guid incurs only a slight increase in wall-clock time, and its memory consumption remains identical to that of Heun's method.

Table 5: **Values of the guidance magnitude hyperparameter** $w_{\mathrm{stiff}}$.

| Dataset | ImageNet $64\times64$ | | | FFHQ $64\times64$ | | |
|---|---|---|---|---|---|---|
| NFEs | 6 | 8 | 10 | 6 | 8 | 10 |
| Heun | 0.75 | 1.25 | 1.0 | 0.5 | 1.25 | 1.5 |
| DPM-Solver | 1.25 | 1.25 | 1.25 | 1.25 | 1.25 | 1.25 |
| DEIS | 2.5 | 2.5 | 2.5 | 3.5 | 3.5 | 3.5 |

### B.3   CHOICE OF SCALING FUNCTION

We were concerned about excessive amplification from the original factor $\alpha(z)$, whose exponential growth makes it highly sensitive to estimation errors. To mitigate this issue, we consider replacing $\alpha(z)$ with lower-growth alternatives derived from its local behavior around $z = 0$. Specifically, we use the lowest-order nontrivial term of its Taylor expansion, yielding a *quadratic* scaling $z^2$, as well as an even more conservative alternative, $|z|$, which is symmetric and linear in the magnitude of $z$ (See Figure 6a).

In Figure 6b, we present the corresponding ablation results in terms of the FID–FD-DINOv2 trade-off. Compared to the original $\alpha(z)$, both alternatives improve stability by suppressing excessive growth. Among them, $z^2$ consistently achieves lower FD-DINOv2 at comparable or better FID, providing the best trade-off between fidelity and stability. In contrast, $\alpha(z)$ often degrades FD-DINOv2 due to amplification-induced instability, while $|z|$ remains stable but yields more limited gains. These results indicate that the quadratic form $z^2$ strikes a favorable balance by preserving the local behavior of $\alpha(z)$ while avoiding its detrimental exponential growth.

### B.4   GUIDANCE INTEGRATION WITH CFG AND AUTOGUIDANCE

In the main paper Section 5.3, we introduce guidance compatibility of ERK-Guid and ERK-Proj. We define the standard model-based guidance term, which applies to methods such as CFG and Autoguidance, as follows

$$\boldsymbol{g} := \boldsymbol{f}_{\mathrm{main}}(\mathbf{x}_{\sigma_i}; \sigma_i) - \boldsymbol{f}_{\mathrm{guiding}}(\mathbf{x}_{\sigma_i}; \sigma_i). \tag{54}$$

To incorporate ERK-Guid, we simply replace the original drift $\boldsymbol{f}$ with the guided drift as follows

$$\boldsymbol{f}^w(\mathbf{x}_{\sigma_i}; \sigma_i) := \boldsymbol{f}_{\mathrm{main}}(\mathbf{x}_{\sigma_i}; \sigma_i) + (w-1)\boldsymbol{g}, \tag{55}$$

Table 6: **Wall-clock time (seconds per image) on a single RTX 3090 GPU.**

| Method | Avg | Min | Max |
|---|---|---|---|
| Heun | 2.777 | 2.775 | 2.782 |
| **ERK-Guid** (Ours) | 2.794 | 2.785 | 2.811 |

Table 7: **Memory consumption for generating a single image.**

| Method | Avg (MB) |
|---|---|
| Heun | 1906.82 |
| **ERK-Guid** (Ours) | 1906.82 |

Table 8: **Quantitative results of ERK-proj.**

| #step | Method | FD-DINOv2 ↓ | FID ↓ | Precision ↑ | Recall ↑ | IS ↑ |
|---|---|---|---|---|---|---|
| | Autoguidance | 50.4 | **1.36** | 0.691 | 0.628 | 262 |
| 32 | **+ERK-Guid** | 47.6 | **1.36** | 0.694 | **0.630** | 268 |
| | **+ERK-Proj** | **44.9** | **1.36** | **0.710** | 0.605 | **274** |

where $w$ is the scaling hyperparameter. Then, we compute the eigenvector estimator and stiffness estimator as follows:

$$\hat{\mathbf{v}}_{\sigma_i}^w := \frac{\boldsymbol{f}^w\big(\mathbf{x}_{\sigma_i}^{\text{Heun}}; \sigma_i\big) - \boldsymbol{f}^w\big(\mathbf{x}_{\sigma_i}^{\text{Euler}}; \sigma_i\big)}{\big\|\boldsymbol{f}^w\big(\mathbf{x}_{\sigma_i}^{\text{Heun}}; \sigma_i\big) - \boldsymbol{f}^w\big(\mathbf{x}_{\sigma_i}^{\text{Euler}}; \sigma_i\big)\big\|_2}, \tag{56}$$

$$\hat{\rho}_{\sigma_i}^w := \frac{\big\|\boldsymbol{f}^w\big(\mathbf{x}_{\sigma_i}^{\text{Heun}}; \sigma_i\big) - \boldsymbol{f}^w\big(\mathbf{x}_{\sigma_i}^{\text{Euler}}; \sigma_i\big)\big\|_2}{\big\|\mathbf{x}_{\sigma_i}^{\text{Heun}} - \mathbf{x}_{\sigma_i}^{\text{Euler}}\big\|_2}. \tag{57}$$

Finally, we constitute our ERK-Guid as follows

$$\hat{\mathbf{x}}_{\sigma_{i+1}}^{\text{Heun}} = \mathbf{x}_{\sigma_{i+1}}^{\text{Heun}} - h\,\beta\,z^2 \big\langle \boldsymbol{f}_{\sigma_i}^w, \hat{\mathbf{v}}_{\sigma_i}^w \big\rangle \hat{\mathbf{v}}_{\sigma_i}^w, \tag{58}$$

where $\beta = \mathbb{1}_{\{\hat{\rho}_{\sigma_i}^w > w_{\text{con}}\}}$ and $z = w_{\text{stiff}}\,h\,\hat{\rho}_{\sigma_i}^w$.

**Lightweight Extension of Guidance Integration.** We additionally introduce ERK-Proj, which interpolates between model-error–based and LTE-based correction signals, aiming to reduce both errors simultaneously. With $\mathbf{g}$ in Eq. 54, ERK-Proj is defined as follows

$$\eta := e^{-w_{\text{stiff}}\hat{\rho}_{\text{stiff}}}, \tag{59}$$

$$\hat{\mathbf{g}} := \eta\mathbf{g} + \big(1 - \eta\big) \langle \mathbf{g}, \hat{\mathbf{v}}_{\sigma_i} \rangle \hat{\mathbf{v}}_{\sigma_i}, \tag{60}$$

$$\boldsymbol{f}^w(\mathbf{x}_{\sigma_i}; \sigma_i) := \boldsymbol{f}_{\text{main}}(\mathbf{x}_{\sigma_i}; \sigma_i) + (w - 1)\hat{\mathbf{g}}, \tag{61}$$

where $\eta$ adjusts the guidance scaling by the stiffness estimator to interpolate the two guidance signals. In Table 8, we report quantitative results demonstrating the effect of ERK-Proj when combined with Autoguidance.

Table 9: **Guidance configuration for Heun, DPM-Solver, and DEIS.**

| Solver | Pair of states | $h$ | $\beta$ |
|---|---|---|---|
| Heun | $\mathbf{x}_{\sigma_i}, \mathbf{x}_{\sigma_i}^{\text{Euler}}$ | $\sigma_i - \sigma_{i+1}$ | $\{0, 1\}$ |
| DPM-Solver (2S) | $\mathbf{x}_{\sigma_i+\delta}, \mathbf{x}_{\sigma_i}$ | $\sigma_i - \sigma_{i+1}$ | $\{0, 1\}$ |
| DEIS | $\mathbf{x}_{\sigma_i}, \mathbf{x}_{\sigma_{i-1}}$ | $\sigma_{i-1} - \sigma_i$ | $\{0, -1\}$ |

### B.5 PLUG-AND-PLAY MODULE FOR ADVANCED SOLVERS

In the main paper Section 5.3, we present additional experimental results that confirm the effectiveness of our method combined with various solvers.

$$\mathbf{x}_{\sigma_{i+1}}^{\text{solver}} = \text{solver}(\mathbf{x}_{\sigma_i}, \sigma_i, \boldsymbol{f}) \tag{62}$$

$$\hat{\mathbf{x}}_{\sigma_{i+1}}^{\text{solver}} = \mathbf{x}_{\sigma_{i+1}}^{\text{solver}} - h\beta z^2 \langle \boldsymbol{f}(\mathbf{x}_{\sigma_i}, \sigma_i), \hat{\mathbf{v}}_{\sigma_i} \rangle \hat{\mathbf{v}}_{\sigma_i} \tag{63}$$

DPM-Solver (2S) Lu et al. (2022) computes an intermediate state during its two-stage update. We denote this intermediate state as $\mathbf{x}_{\sigma_i+\delta}$, and construct the pair as $\{\mathbf{x}_{\sigma_i+\delta}, \mathbf{x}_{\sigma_i}\}$. DEIS Zhang & Chen (2023) computes its update using previous states in a multi-step formulation. We use the most recent previous state $\mathbf{x}_{\sigma_{i-1}}$ to construct the pair as $\{\mathbf{x}_{\sigma_i}, \mathbf{x}_{\sigma_{i-1}}\}$.

Table 10: **Evaluation of adaptive step-size control under different stiffness thresholds.** (A) ERK-Guid and (B) Heun do not adopt adaptive step-size. (C–G) adapt step-size using thresholds $\tau = 0.5, 1, 2, 5, 10$.

| | Adaptive step-size | Threshold | NFE (Avg.) ↓ | FD-DINOv2↓ | FID ↓ |
|---|---|---|---|---|---|
| (A) **ERK-Guid** (Ours) | ✗ | – | **63** | **86.2** | **2.56** |
| (B) Heun | ✗ | – | **63** | 90.1 | 2.58 |
| (C) | ✓ | $\tau = 0.5$ | 90.7 | 88.9 | 2.57 |
| (D) | ✓ | $\tau = 1.0$ | 85.5 | 89.4 | 2.57 |
| (E) | ✓ | $\tau = 2.0$ | 79.5 | 90.0 | 2.58 |
| (F) | ✓ | $\tau = 5.0$ | 67.6 | 90.1 | 2.58 |
| (G) | ✓ | $\tau = 10.0$ | 64.5 | 90.1 | 2.58 |

Table 11: **Quantitative results of predictor-corrector and ERK-Guid (Ours).**

| Method | Stochasticity | $r$ | NFE | FD-DINOv2 ↓ | FID ↓ |
|---|---|---|---|---|---|
| | ✓ | 0.05 | 64 | 91.5 | 2.65 |
| | ✓ | 0.10 | 64 | 93.0 | 2.66 |
| Predictor-Corector | ✓ | 0.15 | 64 | 98.9 | 2.90 |
| Song et al. (2021b) | ✗ | 0.01 | 64 | 90.6 | 2.61 |
| | ✗ | 0.02 | 64 | 87.4 | 2.73 |
| | ✗ | 0.03 | 64 | 88.6 | 3.33 |
| ERK-Guid (**Ours**) | ✗ | - | **63** | **83.7** | **2.60** |

### B.6 COMPARISON WITH ADAPTIVE STEP SIZE

In Table 10, we analyze our proposed method with a classical adoption of stiffness during diffusion sampling. For the experimental setting, we utilize a pre-trained EDM2 network on ImagNet-512. We use the same EDM discretization and halve the step size whenever the stiffness estimator exceeds thresholds of 0.5, 1, 2, 5, or 10. Although a small threshold (e.g., $\tau = 0.5$) produced modest gains in FID and FD-DINOv2, it requires 1.44× more NFEs than ERK-Guid, making it substantially less efficient. Our method still achieved the best performance and efficiency compared to the baseline with various adaptive step-size.

### B.7 COMPARISON WITH THE PREDICTOR–CORRECTOR SAMPLER

We evaluate the predictor–corrector (PC) sampler from Song et al. (2021b) by applying its corrector step after each Heun update. Since the corrector in Song et al. (2021b) is designed for the reverse SDE rather than an ODE, we also consider a deterministic variant obtained by removing its noise term. We vary the hyperparameter $r$ and measure FID and FD-DINOv2 under a comparable NFE. In Table 11, ERK-Guid achieves strong performance compared to both variants of the PC sampler. The stochastic PC sampler consistently degrades performance across both metrics. In contrast, the deterministic PC sampler of Song et al. (2021b) exhibits a clear trade-off: as the correction scale $r$ increases, FD-DINOv2 increases while FID decreases.

## C ALGORITHM

In the main paper Section 4.3, we introduce our ERK-Guid framework. To provide a clearer step-by-step overview, we outline the full procedure in Algorithm 1. Note that the first sampling step does not admit this guidance mechanism, as it requires an ERK pair from the previous iteration. Therefore, we simply skip guidance at this initial step, which is practically justified since stiffness at initialization is typically very small (i.e., $\beta \approx 0$).

## D LLM USAGE

We only used a large language model as a writing assistant to refine phrasing, grammar, and clarity. It was not used for technical content, experiments, analyses, or results.

---

**Algorithm 1** Sampling procedure with **ERK-Guid**

---

1: **procedure** OURS($\boldsymbol{f}_\theta(\mathbf{x};\sigma), \{\sigma_i\}_{i=0,\dots,N}, w_{\text{stiff}}, w_{\text{con}}, \epsilon$)
2:     sample $\mathbf{x}_0 \sim \mathcal{N}(\mathbf{0}, \sigma_0^2 \mathbf{I}) \in \mathbb{R}^{H \times W \times C}$
3:     **for** $i \leftarrow 0$ to $N-1$ **do**
4:        $\boldsymbol{f}_i \leftarrow \boldsymbol{f}_\theta(\mathbf{x}_i; \sigma_i)$
5:        $h \leftarrow \sigma_i - \sigma_{i+1}$
6:        **if** $i \neq 0$ **then**
7:           $\Delta^{\mathbf{f}} \leftarrow \boldsymbol{f}_i - \boldsymbol{f}_i^{\text{Euler}}$
8:           $\Delta^{\mathbf{x}} \leftarrow \mathbf{x}_i - \mathbf{x}_i^{\text{Euler}}$
9:           $\hat{\mathbf{v}}_i, \hat{\rho} \leftarrow \frac{\Delta^{\mathbf{f}}}{||\Delta^{\mathbf{f}}||}, \frac{||\Delta^{\mathbf{f}}||}{||\Delta^{\mathbf{x}}||}$
10:         $\beta, z \leftarrow (\hat{\rho} > w_{\text{con}}), w_{\text{stiff}} h \hat{\rho}$
11:         $\mathbf{g}_i \leftarrow \beta z^2 (\boldsymbol{f}_i \cdot \hat{\mathbf{v}}_i) \hat{\mathbf{v}}_i$
12:        **else**
13:           $\mathbf{g}_i \leftarrow \mathbf{0}$
14:        **end if**
15:        $\mathbf{x}_{i+1}^{\text{Euler}} \leftarrow \mathbf{x}_i - h \boldsymbol{f}_i$
16:        **if** $i \neq N$ **then**
17:           $\boldsymbol{f}_{i+1}^{\text{Euler}} \leftarrow \boldsymbol{f}_\theta(\mathbf{x}_{i+1}^{\text{Euler}}; \sigma_{i+1})$
18:           $\mathbf{x}_{i+1}^{\text{Heun}} \leftarrow \mathbf{x}_i - h \left(\frac{1}{2} \boldsymbol{f}_i + \frac{1}{2} \boldsymbol{f}_{i+1}^{\text{Euler}}\right)$
19:           $\mathbf{x}_{i+1} \leftarrow \mathbf{x}_{i+1}^{\text{Heun}} - h \mathbf{g}_i$
20:           $\mathbf{Q} \xleftarrow{\text{buffer}} \mathbf{x}_{i+1}^{\text{Euler}}, \boldsymbol{f}_{i+1}^{\text{Euler}}$
21:        **else**
22:           $\mathbf{x}_{i+1} \leftarrow \mathbf{x}_{i+1}^{\text{Euler}}$
23:        **end if**
24:     **end for**
25:     **return** $\mathbf{x}_N$
26: **end procedure**

---

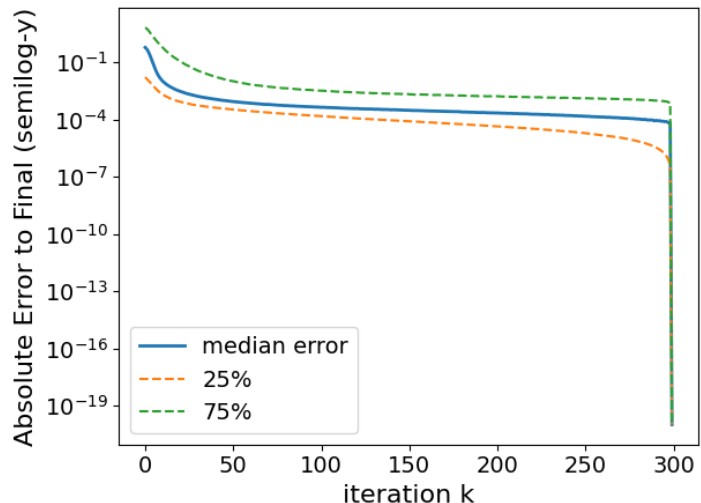

Figure 7: **Convergence of stiffness estimation under JVP power iteration.** The plot shows the absolute error between the estimated stiffness at iteration k and the final converged value. The solid line denotes the median across all seeds and timesteps, while the dashed lines indicate the 25th and 75th percentiles. Errors decrease rapidly and stabilize, demonstrating reliable convergence of the iteration procedure.

$w_{stiff}$

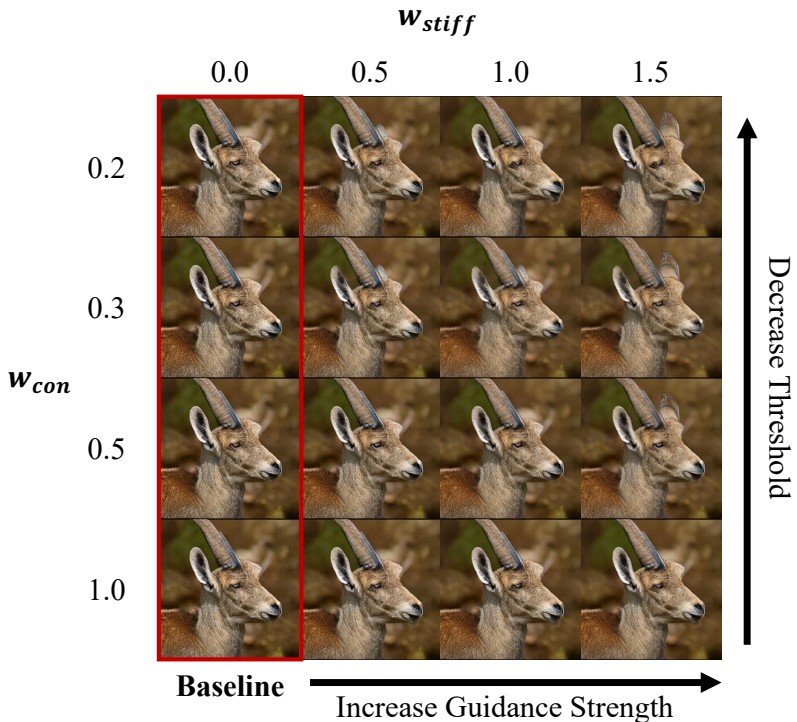

Figure 8: **Qualitative results of ERK-Guid across adjusting guidance scale.**

## E QUALITATIVE RESULTS

In this section, we present qualitative results of ERK-Guid on ImageNet $512 \times 512$. Figures 8 and 9 show that applying ERK-Guid enhances image fidelity when sufficient guidance is applied. These results demonstrate that our method effectively mitigates solver-induced errors during conditional updates along the sampling trajectory.

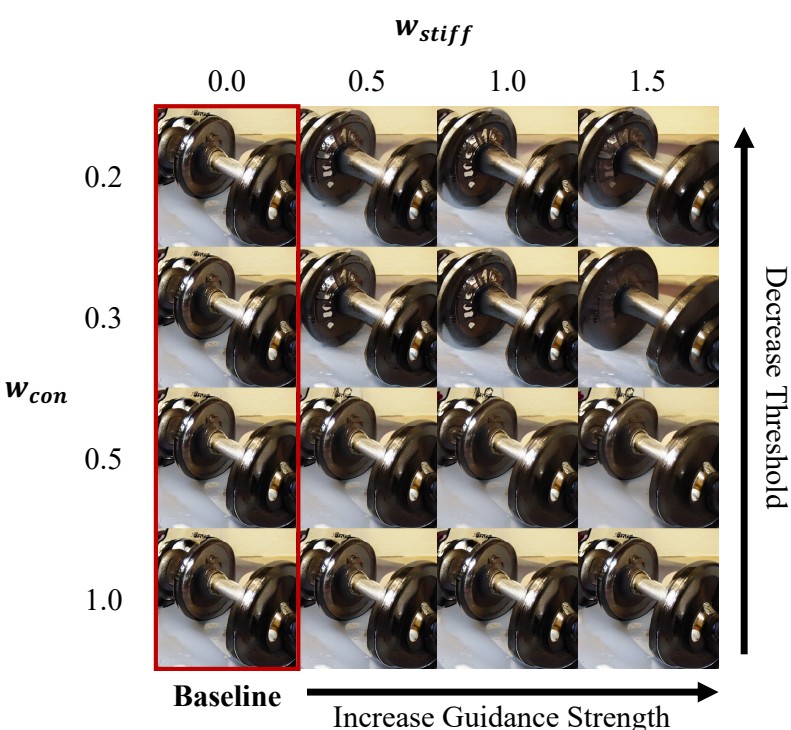

Figure 9: **Qualitative results of ERK-Guid across guidance scales.**

