# OpenReview forum: "Error as Signal: Stiffness-Aware Diffusion Sampling via Embedded Runge-Kutta Guidance"
_ICLR.cc/2026/Conference — ICLR 2026 Poster_

### Official Review · Reviewer_Rzwt · 2025-10-21

**Soundness:** 3
**Presentation:** 4
**Contribution:** 3
**Rating:** 4
**Confidence:** 3

**Summary:**

Classifier-Free Guidance (CFG) has established the foundation for guidance mechanisms in diffusion models, showing that well-designed guidance proxies significantly improve conditional generation and sample quality. Autoguidance (AG) has extended this idea, but it relies on an auxiliary network and leaves solver-induced errors unaddressed. In stiff regions, the ODE trajectory changes sharply, where local truncation error (LTE) becomes a critical factor that deteriorates sample quality. This paper's key observation is that these errors align with the dominant eigenvector, motivating it to target the solver-induced error as a guidance signal. This paper proposes \textbf{E}mbedded \textbf{R}unge–\textbf{K}utta based \textbf{Guid}ance (ERK-Guid), which exploits detected stiffness to reduce LTE and stabilize sampling. This paper theoretically and empirically analyzes stiffness and eigenvector estimators with solver errors to motivate the design of ERK-Guid. This paper's experiments on both synthetic datasets and the popular benchmark dataset ImageNet demonstrate that ERK-Guid consistently outperforms state-of-the-art methods.

**Strengths:**

1. Unlike Autoguidance (AG) which relies on an auxiliary network and fails to address solver-induced errors, ERK-Guid avoids dependence on auxiliary networks and specifically targets solver-induced errors, overcoming key drawbacks of prior guidance mechanisms.

2. Leverages a key observation that solver-induced errors align with the dominant eigenvector, innovatively using such errors as a guidance signal—providing a novel direction for optimizing diffusion model sampling.

3. Exploits detected stiffness in ODE trajectories (where trajectories change sharply) to reduce local truncation error (LTE) and stabilize sampling, directly mitigating a critical factor that degrades sample quality in stiff regions.

**Weaknesses:**

1. Lack of visualization comparison.

2. More experiments are needed. e.g. t2i, t2v.

3. Missing related works with bespoke solver[1, 2, 3], which also searches the optimal solver parameters (linear multisteps solver, RK solver) of a pretrained diffusion model.

[1] Xue, Shuchen, et al. "Accelerating diffusion sampling with optimized time steps." Proceedings of the IEEE/CVF Conference on Computer Vision and Pattern Recognition. 2024.

[2] Wang, Shuai, et al. "Differentiable Solver Search for Fast Diffusion Sampling."  International Conference on Machine Learning (ICML) 2025.

[3] Shaul, Neta, et al. "Bespoke solvers for generative flow models." arXiv preprint arXiv:2310.19075 (2023).

**Questions:**

In practice, extrapolation acceleration is more widely used in sampling. Could this method apply to linear multistep solvers (Adams–Bashforth solver)?

---

> ### Author Response · Authors · 2025-11-23
>
> > ***Q3. Missing related works with bespoke solver[1, 2, 3], which also searches the optimal solver parameters (linear multisteps solver, RK solver) of a pretrained diffusion model.***
> >
> We apologize for the confusion. ERK-Guid is not a bespoke solver but a guidance module that leaves the underlying solver unchanged. Although we do not compare against bespoke solvers, ERK-Guid can be applied as a plug-and-play module on top of advanced solver methods such as DPM-Solver [4] and DEIS [5]. We have also added bespoke solver approaches you mentioned to Related Works. Please refer the reviewer to the ***Global Response*** for further clarification.
>
> [4] Lu, Cheng, et al. "Dpm-solver: A fast ode solver for diffusion probabilistic model sampling in around 10 steps." *NeurIPS* (2022).
>
> [5] Zhang, Qinsheng, and Yongxin Chen. "Fast sampling of diffusion models with exponential integrator." ICLR (2023).
>
> > ***Questions. In practice, extrapolation acceleration is more widely used in sampling. Could this method apply to linear multistep solvers (Adams–Bashforth solver)?***
> >
>
> Our method is compatible with extrapolation-based acceleration. In fact, ERK-Guid has already been applied as a plug-and-play module on top of higher-order solvers. Please refer to the ***Global Response* (c)** for further clarification. We provide additional experiments built on solver approaches.

---

> ### Author Response · Authors · 2025-11-26
>
> > ***Q1. Lack of visualization comparison.***
> >
>
> Thank you for the helpful suggestion. We have revised the paper and added new qualitative comparisons in Figure 6.
>
> In Section 5.4 of the main paper, we evaluate our plug-and-play adaptability across various solver methods. To further demonstrate the effectiveness and architectural generalizability of ERK-Guid, we additionally extend our evaluation to Diffusion Transformer (DiT) [6] based text-to-image (T2I) generation using PixArt-$\alpha$ [7], as shown in Figure 6.
>
> > ***Q2. More experiments are needed. e.g. t2i, t2v.***
> >
>
> We conducted four additional experiments to address this concern:
>
> **(1) Plug-and-play adaptation for image generation (Section 5.4).**
>
> As discussed in the Global Response, we evaluate ERK-Guid on ImageNet-64 and FFHQ-64 with three different solvers, consistently demonstrating strong guidance and correction performance.
>
> **(2) Text-to-Image (T2I) generation using PixArt-$\alpha$.**
>
> Beyond EDM2, we validate ERK-Guid on the DiT-based PixArt-$\alpha$ [7] model for T2I generation, confirming both applicability and architectural generalizability.
>
> **(3) Adaptive step-size evaluation** (Appendix B.5 and Table 8)**.**
>
> We additionally evaluate classical stiffness-based step-size adaptation and show that ERK-Guid remains the most effective and efficient method.
>
> **(4) Comparison with the Predictor–Corrector scheme** (Appendix B.6 and Table 9)**.**
>
> We additionally compare against the PC sampler [8] and demonstrate that ERK-Guid outperforms both the stochastic and deterministic variants across metrics.
>
> We hope that our responses to **Q1, Q2, and Q3**, along with the newly added experiments and qualitative results, fully address the reviewer’s concerns. We would be grateful for any further feedback.
>
> [6] Peebles, William, and Saining Xie. "Scalable diffusion models with transformers." ICCV 2023.
>
> [7] Chen, Junsong, et al. "Pixart-$\alpha$: Fast training of diffusion transformer for photorealistic text-to-image synthesis." ICLR 2024
>
> [8] Song et al. Score-Based Generative Modeling through Stochastic Differential Equations. ICLR 2021.

---

### Official Review · Reviewer_4VrJ · 2025-10-30

**Soundness:** 3
**Presentation:** 3
**Contribution:** 2
**Rating:** 4
**Confidence:** 3

**Summary:**

Based on the situation that LTE in stiff regions aligns with the dominant eigenvector of the drift function’s Jacobian, this paper proposes Embedded Runge–Kutta Guidance is a new guidance method for diffusion models. The experiments validate ERK-Guid's effectiveness and can combine well with existing guidance methods.

**Strengths:**

- The paper rigorously derives the alignment between LTE/dominant eigenvectors via local linearization of the drift field, and proves the approximation accuracy of the stiffness estimator
- This paper designs Comprehensive experiments. Evaluations cover synthetic data and ImageNet with metrics including fidelity, diversity, and alignment. This paper also combines ERK-Guid with existing guidance methods and these methods don't conflict with each other.
- ERK-Guid improves quality in low-step regimes, which means it has the potential to do acceleration in other works.

**Weaknesses:**

- Experiments are limited to EDM2 and ImageNet. It remains unproven whether ERK-Guid works for other architectures or non-image tasks. Stiffness varies across tasks, so generalization needs more validation.
- The performance improvement is not significant, and it is hard to determine whether it is superior to the existing methods. The most crucial point is that even after combining ERK-Guid with other guiding methods, degradation occurred on some datasets, which requires thorough argumentation.

**Questions:**

- See weaknesses.
- Typo should also be considered. For example, the first row of Table 2 and Table 3 has the same content but different formats.
- Please make sure to use vector graphics.

---

> ### Author Response · Authors · 2025-11-23
>
> > ***Q1. Experiments are limited to EDM2 and ImageNet. It remains unproven whether ERK-Guid works for other architectures or non-image tasks. Stiffness varies across tasks, so generalization needs more validation.***
> >
>
> We appreciate the reviewer’s constructive suggestion. In addition to ImageNet-512, we evaluated ERK-Guid on FFHQ and ImageNet-64×64, and observed consistent improvements when combined with higher-order solvers. Importantly, **ERK-Guid is a plug-and-play correction module**: it operates on solver-internal signals without modifying the underlying architecture or objective. This design makes the method inherently **generalizable** and allows it to be seamlessly integrated into diverse sampling schemes, including diffusion samplers based on higher-order methods. As noted in the ***Global Response***, this solver-agnostic structure supports broader applicability beyond the specific settings evaluated in this paper.
>
> > ***Q2. The performance improvement is not significant, and it is hard to determine whether it is superior to the existing methods. The most crucial point is that even after combining ERK-Guid with other guiding methods, degradation occurred on some datasets, which requires thorough argumentation.***
> >
>
> Table 3 in main paper, after combining our method, the performance gain is not large, because we conduct the saturated setting of CFG and Autoguidance. Therefore, we validate our guidance scheme under lower sampling steps (e.g., 16). Combining our method incurs significant gain.
>
> | method (16 steps) | FD-DINOv2 | FID | Precision | Recall | IS |
> | :---: | :---: | :---: | :---: | :---: | :---: |
> | CFG (baseline) | 133.89 | 3.60 | 0.593 | 0.673 | 210 |
> | w_stiff : 0.75 | **125.57** | **3.20** | **0.605** | 0.673 | **215** |
> | AG (baseline) | 82.13 | 2.31 | 0.652 | 0.640 | 230 |
> | w_stiff : 0.75 | **75.16** | **1.92** | **0.669** | **0.643** | **236** |
>
> > ***Questions. Typo should also be considered. For example, the first row of Table 2 and Table 3 has the same content but different formats.***
> >
>
> We apologize for the typo. It has been corrected in the revised manuscript, as noted in the Global Response.

---

> ### Author Response · Authors · 2025-11-26
>
> > ***Q1. Experiments are limited to EDM2 and ImageNet. It remains unproven whether ERK-Guid works for other architectures or non-image tasks. Stiffness varies across tasks, so generalization needs more validation.***
> >
>
> We would also like to share that we have conducted additional experiments. Beyond the plug-and-play correction module discussed earlier, we further demonstrate the generalizability of ERK-Guid by evaluating it on the Diffusion Transformer [1]–based architecture PixArt-$\alpha$ [2] for text-to-image generation. The corresponding qualitative results have been added to Figure 6 in the main paper.
>
> We hope these revisions help alleviate your concerns, and we would greatly appreciate any further feedback.
>
> [1] Peebles, William, and Saining Xie. "Scalable diffusion models with transformers." ICCV 2023.
>
> [2] Chen, Junsong, et al. "Pixart-$\alpha$: Fast training of diffusion transformer for photorealistic text-to-image synthesis." ICLR 2024

---

### Official Review · Reviewer_2Zyk · 2025-10-31

**Soundness:** 3
**Presentation:** 3
**Contribution:** 3
**Rating:** 6
**Confidence:** 3

**Summary:**

This paper presents ERK-Guid, a guidance mechanism for diffusion models aimed at reducing the local truncation error (LTE) of ODE solvers, with explicit consideration of the stiffness of ODE trajectories.
The authors provide a mathematical analysis of LTE and embedded Runge-Kutta pairs, leveraging the Jacobian of the score function.
Based on this analysis, they propose cost-free estimators for ODE stiffness and the direction in which LTE is dominant, and demonstrate how to guide the Heun sampler to achieve smaller LTE.
Through experiments on a toy dataset, mathematical observations are confirmed. Further experiments using EDM2 on ImageNet examine design choices for the cost-free estimators and demonstrate the effectiveness of ERK-Guid.

**Strengths:**

1. The theoretical analysis in Section 4 and the toy example provide clear and insightful explanations of the proposed method.
2. The proposed method is supported by solid mathematical justification and ablation studies.
3. The flow of the paper after the introduction is clear and easy to follow, with well-defined mathematical formulations and illustrative figures.
4. The method can be combined with other guidance techniques such as CFG and Autoguidance, suggesting broad applicability.
5. The experimental results demonstrate the effectiveness of ERK-Guid.

**Weaknesses:**

1. The introduction is somewhat difficult to follow, especially regarding the relationship between stiffness and other guidance methods such as CFG and Autoguidance, since the motivation for the proposed method and those for CFG and Autoguidance seem different.
2. It is unclear whether comparing ERK drift and the dominant eigenvector in Figure 2(c) with CFG and Autoguidance is meaningful, since these methods are not designed to minimize ODE solver step error.
3. The choice of the hyperparameters w_con and w_stiff is important for performance, as shown in Table 2. Hyperparameter search appears necessary, which may be considered a weakness.
4. While the paper focuses on guidance mechanisms, a more thorough comparison with advanced ODE solvers for diffusion models, such as those mentioned in the RELATED WORKS section or other models such as GENIE [Dockhorn et al., 2022], would be valuable.
5. Minor comment: Typo in the caption of Table 2 ("InageNet" should be "ImageNet").

[Dockhorn et al., 2022] Tim Dockhorn, et al. "Genie: Higher-order denoising diffusion solvers." Advances in Neural Information Processing Systems 35 (2022): 30150-30166.

**Questions:**

Regarding Weakness 4: Do you have any results or insights regarding the comparison of your method with advanced ODE solvers for diffusion models?

---

> ### Author Response · Authors · 2025-11-23
>
> > ***Q1. The introduction is somewhat difficult to follow, especially regarding the relationship between stiffness and other guidance methods such as CFG and Autoguidance, since the motivation for the proposed method and those for CFG and Autoguidance seem different.***
> >
>
> As described in the Global Response, we have revised the Introduction to clarify this relationship. By adopting the predictor–corrector perspective, we present guidance methods including CFG, Autoguidance, and our approach as the corrector that steer the solver toward more ideal trajectories. This framing makes the connection between stiffness, guidance signals, and our motivation more coherent.
>
> > ***Q2. It is unclear whether comparing ERK drift and the dominant eigenvector in Figure 2(c) with CFG and Autoguidance is meaningful, since these methods are not designed to minimize ODE solver step error.***
> >
>
> We appreciate the reviewer’s insightful observation. Indeed, CFG and Autoguidance are not intended to minimize ODE solver step error, which precisely motivates our approach. Suboptimal trajectories often arise from both model inaccuracies and solver-induced local truncation error (LTE). ERK-Guid specifically targets the latter by estimating and correcting LTE at each step. In this sense, ERK-Guid complements CFG and Autoguidance, which address different aspects of the sampling process as shown in Table 3 of the main paper. We have clarified this distinction in the revised manuscript.
>
> > ***Q3. The choice of the hyperparameters w_con and w_stiff is important for performance, as shown in Table 2. Hyperparameter search appears necessary, which may be considered a weakness.***
> >
> Thank you for the helpful suggestion. We have added **Figure 5** to the main paper. The figure illustrates the quantitative trends of varying $w_{\mathrm{erk}}$  at fixed $w_{\mathrm{stiff}}$ for different sampling steps (e.g., 16 and 8), revealing the trade-off between FID and FD-DINOv2. We believe this visualization makes the results fully understandable and hope it addresses the reviewer’s concern.
>
> > ***Q4. While the paper focuses on guidance mechanisms, a more thorough comparison with advanced ODE solvers for diffusion models, such as those mentioned in the RELATED WORKS section or other models such as GENIE [Dockhorn et al., 2022], would be valuable.***
> >
>
> Thank you for the helpful suggestion. As detailed in the ***Global Response***, we conducted additional experiments by integrating ERK-Guid as a plug-and-play module into advanced solvers, confirming consistent improvements. Please refer to **Global response (c) Additional experimental results**.
>
> > ***Q5. Minor comment: Typo in the caption of Table 2 ("InageNet" should be "ImageNet").***
> >
>
> We apologize for the typo. It has been corrected in the revised manuscript, as noted in the ***Global Response***.

---

> > ### Comment · Reviewer_2Zyk · 2025-11-26
> >
> > Thank you for your rebuttal. My concerns have been addressed.
> > Especially, the revised introduction better explains the motivation and the position of the proposed method.
> > Additional comparison to the PC sampler in a response to Reviewer qmxA also seems valuable.
> > As a result, I have raised my score.

---

### Official Review · Reviewer_qmxA · 2025-11-03

**Soundness:** 2
**Presentation:** 1
**Contribution:** 2
**Rating:** 4
**Confidence:** 3

**Summary:**

In this paper, the authors propose ERK-Guid, a new sampling algorithm for diffusion ODE samplers that uses the embedded Euler/Heun pair to estimate a local stiffness scalar and approximate the dominant eigenvector direction via the ERK drift difference in order to cancel the local truncation errors. The authors demonstrate the effectiveness of their method on toy examples and ImageNet-512.

**Strengths:**

1. The proposed method is “cost-free” ,i.e. no additional network evaluation is needed.
2. ERK-Guid is complementary to CFG and autoguidance.
3. The sampling recipe is clean and easy to follow.

**Weaknesses:**

1. The paper positions itself in the same line of work as CFG and autoguidance. However, the proposed algorithm resembles more of an advanced solver with a corrector rather than a guidance sampling method. The first two pages of the paper read extremely disconnected to the rest of the paper.
2. Following up on the previous point, for an advanced solver paper, the authors fail to compare their algorithm with other strong solvers like DPM-Solver-v3, UniPC and DEIS, which the authors cite but did not provide any empirical comparison to. Moreover, the authors also cite a classic remedy, adaptive step-size, for stiffness (Petzold (1983); Shampine & Gear (1979)), which they also did not compare in their experiments.
3. Only toy examples and one dataset (Imagenet-512) are examined in their experiments.
4. ERK-Proj, which outperforms the main algorithm ERK-Guid in one of the major experiments, is only introduced on the second to the last paragraph of the main paper with very minimal details in it. There is also very minimal theoretical backing to this algorithm.
5. The theoretical analysis for the entire Section 4.3 is very vague and mostly heuristic. However, it constitutes the algorithms that work in practice.
6. The difference among images in Figure 7 and 8 is extremely subtle and mostly not visible.
7. No wallclock time or memory overhead is compared, which can be a big factor affecting the practicality of the algorithm.

Minor:
    (i) Line 51-52 seems to be missing citations?
    (ii) Line 209 “Let denote”
    (iii) Table 2 caption: “InageNet”

**Questions:**

1. How did the authors determine the hyperparameters $\omega_{conf}$ and $\omega_{stiff}$?

---

> ### Author Response · Authors · 2025-11-23
>
> > ***Q1. The paper positions itself in the same line of work as CFG and autoguidance. However, the proposed algorithm resembles more of an advanced solver with a corrector rather than a guidance sampling method. The first two pages of the paper read extremely disconnected to the rest of the paper.***
> >
>
> We address this concern in the **Global Response**, specifically parts **(a)** and **(b)**.
>
> Part (a) explains the relationship between guidance and solvers under the predictor–corrector view, clarifying that ERK-Guid is a *corrector* Part (b) presents our method in a standard guidance form, showing that it fits naturally within common guidance schemes.
>
> Please refer to **Global Response (a) and (b)** for the detailed explanation.
>
> > ***Q2. Following up on the previous point, for an advanced solver paper, the authors fail to compare their algorithm with other strong solvers like DPM-Solver-v3, UniPC and DEIS, which the authors cite but did not provide any empirical comparison to. Moreover, the authors also cite a classic remedy, adaptive step-size, for stiffness (Petzold (1983); Shampine & Gear (1979)), which they also did not compare in their experiments.***
> >
>
> We appreciate the reviewer’s valuable suggestion regarding empirical comparisons with other solvers. The concern about missing comparisons is fully addressed in **Global Response (c)**, where we provide additional experiments demonstrating that ERK-Guid operates as a plug-and-play correction module for strong solvers. In particular, **Table 1** presents results for Heun, DPM-Solver and DEIS, showing consistent improvements without modifying solver coefficients.
>
> Regarding **adaptive step-size**, using the same EDM discretization, we halved the step size whenever the stiffness estimator exceeded thresholds of 0.5, 1, 2, 5, or 10. Although a small threshold (e.g., τ = 0.5) produced modest gains in FID and FD_DINO, it required **1.44× more NFEs** than ERK-Guid, making it substantially less efficient. Our method still achieved the best performance and efficiency compared to the baseline with various adaptive step-size. These results are added in **Appendix B.5.**
>
> |  | threshold $\tau$ | Average NFE$\downarrow$ | FD-DINOv2 $\downarrow$ | FID $\downarrow$ |
> | --- | --- | :---: | :---: | :---: |
> | w/o | (A) Ours | **63** | **86.2** | **2.56** |
> | w/o | (B) Heun | 63 | 90.1 | 2.58 |
> | w/ | (C) $\tau=0.5$ | 90.7 | 88.9 | 2.57 |
> | w/ | (D) $\tau=1$ | 85.5 | 89.4 | 2.57 |
> | w/ | (E) $\tau=2$ | 79.5 | 90.0 | 2.58 |
> | w/ | (F) $\tau=5$ | 67.6 | 90.1 | 2.58 |
> | w/ | (G) $\tau=10$ | 64.5 | 90.1 | 2.58 |
>
> > ***Q3. Only toy examples and one dataset (Imagenet-512) are examined in their experiments.***
> >
>
> We thank the reviewer for the helpful comment. As summarized in **Global Response (c)**, we expanded our experiments by adding **plug-and-play evaluations on higher-order solvers** (Heun, DPM-Solver, and DEIS) and by testing on additional datasets, including **ImageNet-64 and FFHQ-64**. We also conducted the **adaptive step-size experiment**. These results demonstrate that our method generalizes well across solvers, datasets, and stepping strategies.
>
> > ***Q4. ERK-Proj, which outperforms the main algorithm ERK-Guid in one of the major experiments, is only introduced on the second to the last paragraph of the main paper with very minimal details in it. There is also very minimal theoretical backing to this algorithm.***
> >
>
> Diffusion sampling errors arise from two sources: **solver error (LTE)** and **model error**. ERK-Guid uses LTE as its guidance signal, whereas CFG and Autoguidance uses the discrepancy induced by model error. ERK-Proj is a simple variant to combine our ERK-GUID with Autoguidance via interpolating two guidances.
>
> We provided the more detailed description of ERK-Proj in Appendix B.2, clarifying its formulation and its connection to ERK-Guid. We also updated Table 3 in the main paper with results under lower NFEs for a fair comparison, and added a reference in main paper Section 5.4 to guide readers to the detailed explanation.

---

> ### Author Response · Authors · 2025-11-23
>
> > ***Q5. The theoretical analysis for the entire Section 4.3 is very vague and mostly heuristic. However, it constitutes the algorithms that work in practice.***
> >
> Although the local truncation errors of Heun’s method in Eq. 14 and ERK-Guid in Eq. 21 may appear disconnected, the derivation is straightforwardly obtained via Taylor approximation and the dominant eigen vector.
>
> Let assume $v\_1$ refers **ground truth** dominant eigenvector.
>
> As discussed in Section 4.1, when the dominant eigenvector $v\_1$ governs local dynamics, the LTE of Heun’s method is dominated along the direction of $v\_1$:
>
> $$
> \text{LTE}\^\text{Heun}:=\mathbf{x}\_{\sigma\_{i+1}} -~ \mathbf{x}\^{\text{Heun}}\_{\sigma\_{i+1}}\approx-h\alpha(z\_1)\bigl\langle \mathbf{f}\_{\mathbf{x}\_{\sigma\_i}},\ \mathbf{v}\_1 \bigr\rangle~\mathbf{v}\_1.
> $$
>
> Rearranging terms gives
> $$
> \mathbf{x}\_{\sigma\_{i+1}} \approx\mathbf{x}\^{\text{Heun}}\_{\sigma\_{i+1}}-h\alpha(z\_1)\bigl\langle \mathbf{f}\_{\mathbf{x}\_{\sigma\_i}},\ \mathbf{v}\_1 \bigr\rangle~\mathbf{v}\_1.
> $$
>
> Because $\alpha(z)=\frac{e\^z-1}{z}-1-\frac 1 2z$, its Taylor expansion at $z=0$ yields
> $$
> \alpha(z\_1)=\frac 1 6 z\_1\^2+O(z\_1\^3),
> $$
> and substituting this Taylor approximation into Eq.14 gives
> $$
> \mathbf{x}\_{\sigma\_{i+1}}\approx\mathbf{x}\_{\sigma\_{i+1}}\^{\mathrm{Heun}}-\frac 1 6 hz\_1\^2\big\langle \mathbf{f}\_{\mathbf{x}\_{\sigma\_i}},{\mathbf{v}}\_{1} \big\rangle{\mathbf{v}}\_{\_1}.
> $$
>
> We interpret this additive term as a guidance correction and introduce a tunable scale $w\_{\text{stiff}}$ as follows
> $$
> \hat{\mathbf{x}}\_{\sigma\_{i+1}}\^{\mathrm{Heun}}=\mathbf{x}\_{\sigma\_{i+1}}\^{\mathrm{Heun}}-hz\^2\big\langle \mathbf{f}\_{\mathbf{x}\_{\sigma\_i}},{\mathbf{v}}\_{1} \big\rangle{\mathbf{v}}\_{\_1},
> \quad z:=w\_{\mathrm{stiff}}~z\_1,
> $$
> where constant $\frac 1 6$ is absorbed into $w\_{\text{stiff}}$.
>
> Since neither the dominant eigenvector $v\_1$ nor the eigenvalue $\lambda$ is available in practice, we replace them with our cost-free estimators $\hat{v}\_1$ and stiffness $\hat\rho \approx|\lambda\_1|$. This yields the practical ERK-Guid update:
> $$
> \hat{\mathbf{x}}\_{\sigma\_{i+1}}\^{\mathrm{Heun}}=\mathbf{x}\_{\sigma\_{i+1}}\^{\mathrm{Heun}}-hz\^2\big\langle \mathbf{f}\_{\mathbf{x}\_{\sigma\_i}},\hat{\mathbf{v}}\_{\mathbf{x}\_{\sigma\_i}} \big\rangle\hat{\mathbf{v}}\_{\mathbf{x}\_{\sigma\_i}},
> \quad z\approx w\_{\mathrm{stiff}}~h\hat\rho.
> $$
>
> Since the derivation assumes operation within stiff regions, we introduce a step function $\beta$ that activates the guidance only when the estimated stiffness exceeds a threshold, i.e., $\hat{\rho}>w\_\text{con}$.
> $$
> \hat{\mathbf{x}}\_{\sigma\_{i+1}}\^{\mathrm{Heun}}=\mathbf{x}\_{\sigma\_{i+1}}\^{\mathrm{Heun}}-h\beta z\^2\big\langle \mathbf{f}\_{\mathbf{x}\_{\sigma\_i}},\hat{\mathbf{v}}\_{\mathbf{x}\_{\sigma\_i}}\big\rangle\hat{\mathbf{v}}\_{\mathbf{x}\_{\sigma\_i}},\quad z\approx w\_{\mathrm{stiff}}~h\hat\rho.
> $$
>
> Q.E.D.
>
> Moreover, ERK-Guid (Eq. 21) can be rewritten into a standard guidance form, as shown in **Global Response (b)**. This demonstrates that our method in Section 4.3 is theoretically connected to existing guidance schemes, rather than being heuristic. For a full derivation, please refer to **Global Response (b): Rewritten Proposed Method** or **Appendix A.4**.
>
> $$
> \hat{\mathbf{x}}\_{\sigma\_{i+1}}\^{\mathrm{Heun}}= \mathbf{x}\_{\sigma\_{i+1}}\^{\mathrm{Heun}}+\gamma(\mathbf{x}\_{\sigma\_i},\mathbf{x}\_{\sigma\_i}\^{\mathrm{Euler}},\sigma\_i)
> \bigg(
> \mathbf{f}\left(\mathbf{x}\_{\sigma\_i};\sigma\_i\right)
> -\mathbf{f}\left(\mathbf{x}\_{\sigma\_i}\^{\mathrm{Euler}};\sigma\_i\right)
> \bigg)
> $$
>
> $$
> \gamma(\mathbf{x}\_{\sigma\_i},\mathbf{x}\_{\sigma\_i}\^{\mathrm{Euler}},\sigma\_i)=\frac {-h\ \beta\ z\^2 \big\langle \mathbf{f}\_{\mathbf{x}\_{\sigma\_i}},\hat{\mathbf{v}}\_{\mathbf{x}\_{\sigma\_i}} \big\rangle}{\big\|\mathbf{f}\left(\mathbf{x}\_{\sigma\_i};\sigma\_i\right)
> -\mathbf{f}\left(\mathbf{x}\_{\sigma\_i}\^{\mathrm{Euler}};\sigma\_i\right)\big\|\_2}.
> $$
> > ***Q7. No wallclock time or memory overhead is compared, which can be a big factor affecting the practicality of the algorithm.***
> >
> We evaluate the wall-clock time and memory overhead on the ImageNet 512×512 dataset using a single RTX 3090 GPU with batch size 1, comparing ERK-Guid against Heun’s method. ERK-Guid incurs only a slight increase in wall-clock time, and its memory consumption remains identical to that of Heun’s method.
> | Wall clock (s/image) | | Avg | Min | Max |
> | --- | --- | :---: | :---: | :---: |
> | Huen | |  2.777 | 2.7749 | 2.7818 |
> | ERK-Guid | | 2.794 | 2.7846 | 2.8108 |
>
> | Memory Consumption | | Avg |
> | --- | --- |:---: |
> | Huen | |1906.82 MB |
> | ERK-Guid | | 1906.82 MB |
> > ***Questions. How did the authors determine the hyperparameters $w_{\mathrm{con}}$ and $w_{\mathrm{stiff}}$?***
> >
> We determined $w_{\mathrm{con}}$ and $w_{\mathrm{stiff}}$ using a simple grid search, and the resulting performance trends are shown in the newly added Figure 5.

---

> > ### Comment · Reviewer_qmxA · 2025-11-25
> >
> > Thank you for your rebuttal! This resolves most of my questions and concerns, especially the connection between predictor-corrector and the additional empirical results. I would highly suggest the authors to include these discussions in the final manuscript as they significantly help clarifying the paper.
> >
> > However, I do have one question/concern remains: if the authors agree that the proposed algorithm is a predictor-corrector algorithm, why would't they compare to the original predictor-corrector algorithm proposed in [1]?
> >
> > [1] Song et al. Score-Based Generative Modeling through Stochastic Differential Equations. ICLR 2021.

---

> > > ### Author Response · Authors · 2025-11-26
> > >
> > > We thank the reviewer for the helpful follow-up question.
> > >
> > > We evaluate the predictor–corrector (PC) sampler from [1] by applying its corrector step after each Heun update. Because the corrector step [1] is designed for the reverse SDE rather than an ODE, we also consider a deterministic variant obtained by removing the noise term. We vary the hyperparameter $r$ and measure FID and FD-DINOv2 under a comparable sampling budget to our method (63 vs. 64 NFEs).
> > >
> > > | Method | Stochasticity | | $r$ | NFE | FID | FD-DINOv2 |
> > > | --- | :---: | :---: | :---: | :---: | :---: | :---: |
> > > | Song et al. [1] | O | | 0.05 | 64 | 2.65 | 91.5 |
> > > |  | O | | 0.10 | 64 | 2.66 | 93.0 |
> > > |  | O | | 0.15 | 64 | 2.90 | 98.9 |
> > > |  | X | | 0.01 | 64 | 2.61 | 90.6 |
> > > |  | X | | 0.02 | 64 | 2.73 | 87.4 |
> > > |  | X | | 0.03 | 64 | 3.33 | 88.6 |
> > > | ERK-Guid (**Ours**)  | X | | - | **63** | **2.58** | **83.7** |
> > > | | | | | | | |
> > >
> > > While the original PC sampler [1] either degrades performance or creates a trade-off between the two metrics, ERK-Guid consistently improves both metrics simultaneously and clearly outperforms the original PC sampler.
> > >
> > > Thank you again for the constructive feedback. We include these results in Appendix B.6 and Table 9.
> > >
> > > [1] Song et al. Score-Based Generative Modeling through Stochastic Differential Equations. ICLR 2021.

---

> ### Author Response · Authors · 2025-11-26
>
> > ***Q6. The difference among images in Figure 7 and 8 is extremely subtle and mostly not visible.***
> >
>
> Thank you for the helpful suggestion. We have revised the main paper with newly added qualitative comparisons in Figure 6. (The previous Figure 7 and Figure 8 have been moved to Figure 9 and Figure 10, respectively, and we kindly ask for your understanding regarding this change.)
>
> To further strengthen the plug-and-play adaptability analysis in Section 5.4, we additionally apply our guidance method to DPM-Solver and include high-fidelity generation results. Moreover, to demonstrate architectural generalizability, we extend our evaluation to Diffusion Transformer (DiT) [2]–based text-to-image (T2I) generation using PixArt-$\alpha$ [3], also shown in Figure 6.
>
> In Figure 6, we highlight the blue-boxed regions to more clearly illustrate the qualitative differences.
>
> We hope this revision alleviates the reviewer’s concern. Please refer to Figure 6 for the updated results.
>
> [2] Peebles, William, and Saining Xie. "Scalable diffusion models with transformers." ICCV 2023.
>
> [3] Chen, Junsong, et al. "Pixart-$\alpha$: Fast training of diffusion transformer for photorealistic text-to-image synthesis." ICLR 2024

---

> > ### Comment · Reviewer_qmxA · 2025-11-27
> >
> > Thank you for your response. My concerns are fully addressed so I am willing to increase my score from 4 to 6.

---

### Author Response · Authors · 2025-11-23
**Global reponse**

We thank all reviewers for their constructive feedback and helpful suggestions.

We first briefly explain why our method is “guidance”, by clarifying ***(a) the relationship between guidance and solvers*** using predictor-corrector scheme. And then, we introduce ***(b) the alternative (but equivalent) formulation of our proposed method*** as a common form of guidance schemes. Lastly, we present **(*c) additional experimental results*** that confirm the effectiveness of our method combined with various solvers as a guidance/corrector.

**(a) Relationship between guidance and solvers**

Recent work [1] shows that guidance can be interpreted as a corrector in a predictor–corrector scheme where a solver (predictor) provides the initial predictions and guidance (corrector) adjusts them. In this sense, guidance schemes, including CFG, Autoguidance, and ours, are correctors.

**(b) Alternative (but equivalent) formulation**

To show the resemblance between our ERK-Guid and other guidances, we rewrite Eq. 21 in an alternative form as

$$
\hat{\mathbf{x}}\_{\sigma\_{i+1}}\^{\mathrm{Heun}}= \mathbf{x}\_{\sigma\_{i+1}}\^{\mathrm{Heun}}+ \underbrace{\gamma(\mathbf{x}\_{\sigma\_i},\mathbf{x}\_{\sigma\_i}\^{\mathrm{Euler}},\sigma\_i)}\_{\text{adaptive guidance scaling}}
\bigg(
\underbrace{\mathbf{f}\left(\mathbf{x}\_{\sigma\_i};\sigma\_i\right)
-\mathbf{f}\left(\mathbf{x}\_{\sigma\_i}\^{\mathrm{Euler}};\sigma\_i\right)
}\_{\text{guidance direction}}\bigg).
$$

Our ERK-Guid is a guidance method that leverages the stiffness signal as an adaptive guidance scaling and an guidance direction. The above equation clearly shows that our method is a guidance for diffusion models. Detailed derivation is provided below.


**(c) Additional experimental results**

Finally, similar to other guidance methods, ERK-Guid can be applied to various solvers as a plug-and-play correction module. To demonstrate its effectiveness and broad applicability, we provide additional experimental results in 18 different settings (3 solvers x 2 datasets x 3 NFEs). The experimental results show that our method consistently improves performance and provided up to **34.17 FID improvement** (see, FFHQ (6) with DPM-solver).

- Table 1 (FID $\downarrow$)
| Dataset / NFE | ImageNet (6) | ImageNet (8) | ImageNet (10) |  || FFHQ (6) | FFHQ (8) | FFHQ (10) |
| --- | :---: | :---: | :---: | :---: |:---: | :---: | :---: | :---: |
| Heun | 89.63 | 37.65 | 16.46 |  | | 142.4 | 57.21 | 29.54 |
| Huen + **ERK-Guid** (Ours) | **85.19** | **35.92** | **13.85** |  | | **132.8** | **54.73** | **23.38** |
| DPM-Solver (2S) [2] | 44.83 | 12.42 | 6.84 |  | | 83.17 | 22.84 | 9.46 |
| DPM-Solver (2S) + **ERK-Guid** (Ours) | **31.59** | **10.58** | **6.54** |  | | **49.0** | **10.44** | **4.64** |
| DEIS [3] | 12.57 | 6.84 | 5.34 |  | | 12.25 | 7.59 | 5.56 |
| DEIS + **ERK-Guid** (Ours) | **9.56** | **6.25** | **4.89** |  | | **9.96** | **6.04** | **4.47** |

**Table 1** presents results obtained by applying ERK-Guid to **Heun**, **DPM-Solver** [2], and **DEIS** [3], and In all cases, ERK-Guid is incorporated by simply adding the correction term from Eq. 21 of the main paper to the solver’s predicted state, without modifying any solver coefficients. This unified correction rule allows ERK-Guid to operate seamlessly across diverse higher-order solvers, confirming its solver-agnostic nature.

For implementation details on how ERK-Guid is integrated into each solver, please refer to **Appendix B.4**.


- Major update of manuscript
    - (1) Introduction & Related Works
    - (2) Experiments Section 5.4
    - Appendix: (3) Section A.3, (4) A.4, (5) B.2, (6) B.4, (7) B.5

We hope that these additional explanations and expanded experimental results sufficiently address the reviewers’ concerns regarding both conceptual clarity and experimental completeness.

[1] Bradley, Arwen, and Preetum Nakkiran. "Classifier-free guidance is a predictor-corrector." *NeurIPS Workshop* (2024).

[2] Lu, Cheng, et al. "Dpm-solver: A fast ode solver for diffusion probabilistic model sampling in around 10 steps." *NeurIPS* (2022).

[3] Zhang, Qinsheng, and Yongxin Chen. "Fast sampling of diffusion models with exponential integrator." ICLR (2023).

---

> ### Author Response · Authors · 2025-11-23
>
> **Proof.**
>
> In the main paper (Eq. 21), the ERK\-Guid updates the Huen prediction as
>
> $$
> \hat{\mathbf{x}}\_{\sigma\_{i+1}}\^{\mathrm{Heun}}
> = \mathbf{x}\_{\sigma\_{i+1}}\^{\mathrm{Heun}}
> \-h\beta z\^2
> \big\langle \mathbf{f}\_{\mathbf{x}\_{\sigma\_i}},\hat{\mathbf{v}}\_{\mathbf{x}\_{\sigma\_i}} \big\rangle
> \hat{\mathbf{v}}\_{\mathbf{x}\_{\sigma\_i}},
> $$
>
> where $h =\sigma\_i\-\sigma\_{i+1}$, $\beta := {1}\_{\{\hat{\rho}\_{\mathbf{x}\_{\sigma\_i}} > w\_{\mathrm{con}}\}}$, and $z = w\_{\mathrm{stiff}}h\hat{\rho}\_{\mathbf{x}\_{\sigma\_i}}$.
>
> We abbreviate the drift as $\mathbf{f}\_{\mathbf{x}\_{\sigma\_i}}$ and eigenvector estimator $\hat{\mathbf{v}}\_{\mathbf{x}\_{\sigma\_i}}$, where the latter is defined by Eq. 17:
>
> $$
> \hat{\mathbf{v}}\_{\mathrm{stiff}}(\mathbf{x}\_{\sigma\_i},\sigma\_i)
> :=
> \frac{\mathbf{f}\left(\mathbf{x}\_{\sigma\_i};\sigma\_i\right)
> \- \mathbf{f}\left(\mathbf{x}\_{\sigma\_i}\^{\mathrm{Euler}};\sigma\_i\right)}
> {\big\|\mathbf{f}\left(\mathbf{x}\_{\sigma\_i};\sigma\_i\right)
> \- \mathbf{f}\left(\mathbf{x}\_{\sigma\_i}\^{\mathrm{Euler}};\sigma\_i\right)\big\|\_2}.
> $$
>
> Substituting this expression into Eq. 21 yields
>
> $$
> \hat{\mathbf{x}}\_{\sigma\_{i+1}}\^{\mathrm{Heun}}=
> \mathbf{x}\_{\sigma\_{i+1}}\^{\mathrm{Heun}}
> \-\frac{h\beta z\^2
> \big\langle \mathbf{f}\_{\mathbf{x}\_{\sigma\_i}},\hat{\mathbf{v}}\_{\mathbf{x}\_{\sigma\_i}} \big\rangle}
> {\big\|\mathbf{f}\left(\mathbf{x}\_{\sigma\_i};\sigma\_i\right)
> \-\mathbf{f}\left(\mathbf{x}\_{\sigma\_i}\^{\mathrm{Euler}};\sigma\_i\right)\big\|\_2}\bigg(\mathbf{f}\left(\mathbf{x}\_{\sigma\_i};\sigma\_i\right)
> \-\mathbf{f}\left(\mathbf{x}\_{\sigma\_i}\^{\mathrm{Euler}};\sigma\_i\right)\bigg).
> $$
>
> For simplicity, we introduce an adaptive scaling function
>
> $$
> \gamma(\mathbf{x}\_{\sigma\_i},\mathbf{x}\_{\sigma\_i}\^{\mathrm{Euler}},\sigma\_i)=\frac {\-h\ \beta\ z\^2 \big\langle \mathbf{f}\_{\mathbf{x}\_{\sigma\_i}},\hat{\mathbf{v}}\_{\mathbf{x}\_{\sigma\_i}} \big\rangle}{\big\|\mathbf{f}\left(\mathbf{x}\_{\sigma\_i};\sigma\_i\right)
> \-\mathbf{f}\left(\mathbf{x}\_{\sigma\_i}\^{\mathrm{Euler}};\sigma\_i\right)\big\|\_2}.
> $$
>
> The ERK\-Guid update can then be rewritten as
>
> $$
> \hat{\mathbf{x}}\_{\sigma\_{i+1}}\^{\mathrm{Heun}}= \mathbf{x}\_{\sigma\_{i+1}}\^{\mathrm{Heun}}+ \gamma(\mathbf{x}\_{\sigma\_i},\mathbf{x}\_{\sigma\_i}\^{\mathrm{Euler}},\sigma\_i)
> \bigg(
> \mathbf{f}\left(\mathbf{x}\_{\sigma\_i};\sigma\_i\right)
> \-\mathbf{f}\left(\mathbf{x}\_{\sigma\_i}\^{\mathrm{Euler}};\sigma\_i\right)
> \bigg).
> $$
>
> The above equation clearly shows that our method is a guidance with an adaptive guidance scaling $\gamma$.

---

### Meta-Review · Area_Chair_W4qC · 2026-01-06

**Summary:**

This paper proposes a new sampling algorithm for diffusion models. Reviewers had 2 main complaints:

- A lack of clarity regarding the positioning of the paper, as it is positioned as a guidance work rather than an ODE sampler.

- A lack of thorough experiments, both in terms of datasets and baselines.

The authors provided a comprehensive rebuttal which satisfactorily addressed these concerns. I thus recommend acceptance.

**Reviewer Concerns:**

I believe all concerns were addressed satisfactorily.

**Reviewer Scores:**

Reviewers qmxA and 2Zyk both agreed to raise their scores before the leak. I also believe that reviewer Rzwt would have been likely to raise theirs, since their concerns were the same as those of qmxA and 2Zyk, which were properly addressed in the rebuttal.

---

### Decision · Program_Chairs · 2026-01-26

Accept (Poster)